# PREDICTION WITH EXPERT ADVICE UNDER LOCAL DIFFERENTIAL PRIVACY

**Ben Jacobsen**
Department of Computer Sciences
University of Wisconsin — Madison
bjacobsen3@wisc.edu

**Kassem Fawaz**
Department of Electrical and Computer Engineering
University of Wisconsin — Madison
kfawaz@wisc.edu

## ABSTRACT

We study the classic problem of prediction with expert advice under the constraint of local differential privacy (LDP). In this context, we first show that a classical algorithm naturally satisfies LDP and then design two new algorithms that improve it: RW-AdaBatch and RW-Meta. For RW-AdaBatch, we exploit the limited-switching behavior induced by LDP to provide a novel form of privacy amplification that grows stronger on easier data, analogous to the shuffle model in offline learning. Drawing on the theory of random walks, we prove that this improvement carries essentially no utility cost. For RW-Meta, we develop a general method for privately selecting between experts that are themselves non-trivial learning algorithms, and we show that in the context of LDP this carries no extra privacy cost. In contrast, prior work has only considered data-independent experts. We also derive formal regret bounds that scale inversely with the degree of independence between experts. Our analysis is supplemented by evaluation on real-world data reported by hospitals during the COVID-19 pandemic; RW-Meta outperforms both the classical baseline and a state-of-the-art *central* DP algorithm by 1.5-3$\times$ on the task of predicting which hospital will report the highest density of COVID patients each week.

## 1 INTRODUCTION

Many practical algorithmic and organizational problems, from routing data to allocating scarce resources, involve repeated predictions about the future state of a system. Prediction from Expert Advice (Cesa-Bianchi & Lugosi, 2006) is a flexible framework developed in the online learning literature that can be used to model these problems as an iterative game between a player and the environment: at each time step $t = 1, \ldots, T$, the player receives suggestions from each of $n$ 'experts' (which might correspond to literal human advisors, complex algorithms, or static actions like "always do X") and must decide whose advice to follow. Nature then reveals a vector representing the *gain* associated with each expert's suggested action, and the player receives a reward based on the action they actually took. The player's goal is to maximize their reward over the course of the entire game. [1]

Our focus will be on privacy-critical instances of this problem where expert suggestions correspond to predictions about human behavior, such as medical diagnosis (Morino et al., 2015), human mobility (Xu et al., 2015), residential energy usage Devaine et al. (2013) or public health (Altieri et al., 2021). The motivating concern is that if we optimize purely for accuracy in this context, then our algorithm might inadvertently leak sensitive data through its outputs, similar to batch learning (Shokri et al., 2017; Carlini et al., 2023). Here, 'data' refers to a sequence of vectors representing the gain of each possible action at a single time step, which are sensitive because they measure the accuracy of predictions about human behavior.

The algorithms that we study will all satisfy (non-interactive (Smith et al., 2017)) local differential privacy (LDP) (Yang et al., 2023); we assume that noise is added to each gain vector before it is shared, and our aim is to design new algorithms that use this noisy data as efficiently as possible. In contrast, prior works have almost exclusively approached this problem by designing algorithms that satisfy

---

[1]It is more common to describe the problem in terms of minimizing loss. Using gain instead will be notationally convenient later on, and guarantees about one can be converted to the other by flipping signs.

central DP (CDP) (Jain et al., 2012; Thakurta & Smith, 2013; Agarwal & Singh, 2017; Kairouz et al., 2021; Asi et al., 2023). That is, they assume that sensitive data is shared freely with a trusted central curator, and investigate how the curator can introduce noise into their analysis to protect privacy. We are interested in the local model because it eliminates the need for a trusted curator, which can better enable applications in highly-sensitive domains like those mentioned above. A secondary advantage of LDP that it is easy to implement in practice, requiring only the ability to sample independent randomness wherever data is generated, unlike approaches based on secure aggregation that generally require multiple rounds of interaction and/or the implementation of sophisticated cryptographic algorithms Damie et al. (2025).

The first algorithm one might think of in this setting is to sum all of the noisy gain vectors seen so far and choose the expert that appears to have done the best historically. Indeed, this approach is implicit in Agarwal & Singh (2017, 4.1), and is actually identical to an earlier algorithm from the non-private literature by Devroye et al. (2013). The latter algorithm, which we refer to as RW-FTPL (short for Random-Walk Follow-the-Perturbed-Leader), was designed to switch its prediction only a small number of times in expectation. This concept of limited switching motivated our first contribution:

**Contribution 1: Stability under LDP.** We prove Theorem 1, which provides a tight and easily computable characterization of the switching behavior of RW-FTPL. This allows us to answer questions like "What is the largest integer $B$ such that, with probability at least $1 - \alpha$, RW-FTPL will not change its prediction in the next $B$ steps?". Using this tool, we propose a new algorithm, RW-AdaBatch, which modifies RW-FTPL to adaptively batch data together whenever we can guarantee with high probability this our predictions won't change as a result.

RW-AdaBatch is fully invariant to permutations within certain (random, data-dependent) subsequences of its input, and we develop novel proof techniques to show that this translates to amplified CDP guarantees for its outputs, *a la* the local-to-central amplification of the popular shuffle model Erlingsson et al. (2019). This privacy amplification naturally grows stronger when data is 'easy' in the sense that there is a clear best expert at some time step. By construction, it also produces the same output as RW-FTPL at each time step with high probability, meaning that the only cost we pay for this privacy amplification a small multiplicative increase in worst-case regret bounds that can be controlled as a hyperparameter. The analytic lower bounds that we prove for the degree of amplification have no simple closed form, but they can be computed numerically. We visualize them in Figure 1 alongside empirical Monte Carlo simulations that validate their tightness.

**Contribution 2: Private Learning in Dynamic Environments.** In the non-private literature, 'experts' are often taken to themselves be non-trivial learning algorithms. But in a private context, there is an important gap between simply *identifying* the best expert (e.g. determining that $M$ is a good model for the spread of a disease) and actually acting on their advice (e.g. fitting $M$ to historical data and using its predictions to allocate medical resources). When experts are data-dependent, the second step can itself violate privacy.

Prior work avoids this problem by implicitly assuming that experts are data-independent and therefore consume no privacy budget, but this is a significant limitation in practice. To overcome this limitation, we propose RW-Meta, which employs a highly-efficient noise-sharing scheme to both select between multiple data-dependent experts *and* act on their suggestions at no additional privacy cost beyond that of RW-FTPL. In section 4 we show that this can lead to dramatic improvements in practice, outperforming even CDP baselines by more than $1.5\times$ on a prediction task involving data reported by hospitals during the COVID-19 pandemic. In Theorem 3, we prove formal regret bounds for RW-Meta with respect to the best-data-dependent expert. These bounds scale with the leading eigenvalue of a certain matrix related to the empirical covariance of the experts, which can be thought of as representing the difficulty of the learning task.

## 2 BACKGROUND AND RELATED WORK

### 2.1 RELATED WORK

The study of online learning with DP was initiated in 2010 by the papers of Dwork et al. (2010) and Chan et al. (2011), which examined the problem of privately updating a single counter. Later work has gone on to apply their framework to a wide range of tasks, including more complicated

Table 1: Regret and privacy guarantees for our algorithms and prior work with $T$ time steps and $n$ experts. CDP, GCDP, and GLDP stand for central DP, Gaussian central DP and Gaussian local DP.

| Algorithm | Privacy Guarantee | Asymptotic Regret | Regret Definition |
|---|---|---|---|
| Agarwal & Singh (2017) | $\varepsilon, \delta$-CDP | $\sqrt{T \log n} + \frac{\sqrt{n \log n} \log^{1.5} T}{\varepsilon}$ | Best static action in hindsight |
| RW-FTPL (Devroye et al., 2013) | $\mu$-GLDP | $\frac{\sqrt{T n \log n}}{\mu}$ | Best static action in hindsight |
| **RW-AdaBatch** with parameter $\alpha > 0$ | $\mu$-GLDP; **Amplified GCDP** | $(1 + \alpha) \frac{\sqrt{T n \log n}}{\mu}$ | Best static action in hindsight |
| **RW-Meta** with $m$ learners | $\mu$-GLDP | $\frac{\sqrt{T n m \log m}}{\mu}$ | **Best learner in hindsight** |

statistics (Perrier et al., 2018; Bolot et al., 2013; Wang et al., 2021), online convex optimization (Jain et al., 2012; Kairouz et al., 2021; Choquette-Choo et al., 2023; 2024), and the partial information (or 'bandit') setting (Thakurta & Smith, 2013; Hannun et al., 2019; Azize & Basu, 2024). Despite the advantages of LDP described in the introduction, we are only aware of a single existing LDP experts algorithm, by Gao et al. (2024). While their work shares motivation with ours, the federated context and stochastic adversaries they consider lead to substantial technical differences in our results.

Within the central model, the works most similar to ours are Asi et al. (2023; 2024) and Agarwal & Singh (2017). Like us, Asi et al. (2023) exploit the limited-switching behavior of a classical online learning algorithm [2] to develop private experts algorithms. Their focus is on the high-dimensional regime, where the gap between local and central DP is most pronounced (Edmonds et al., 2019; Duchi et al., 2013). Their work has gone on to inspire a flurry of work establishing connections between privacy and limited switching for both the experts problem and online convex optimization (Asi et al., 2024; Agarwal et al., 2023a;b), which our RW-AdaBatch algorithm contributes to.

In a contemporaneous work, Saha et al. (2025) extend Asi et al. (2023) to propose new algorithms for expert advice in dynamic environments, motivated by the same concerns that led us to design RW-Meta. The algorithm for oblivious adversaries in Saha et al. (2025) works by constructing an exponentially large set of candidate experts to pass to the high-dimensional CDP algorithm of Asi et al. (2023). In contrast, the performance guarantees of RW-Meta are defined with respect to a smaller set of carefully chosen candidate experts, leading to a qualitatively different notion of regret and a linear (rather than exponential) runtime dependence on $T$. Because of these computational issues, we do not include their algorithm in our empirical evaluation. Instead, our primary (CDP) baseline is the FTPL algorithm from Agarwal & Singh (2017), which remains state-of-the-art in the low-dimensional regime that we target, where $T \gtrsim n/\varepsilon^2$.

## 2.2 PRELIMINARIES

Given a vector $v \in \mathbb{R}^n$, we denote its $k^{th}$ element with the subscript $v_k$, and its $k^{th}$ *smallest* element using parentheses as in $v_{(k)}$. We use the term 'leader' to refer to the index of $v_{(n)}$, and the term 'gap' to refer to the quantity $v_{(n)} - v_{(n-1)}$. For two random variables $X$ and $Y$, we say that $X$ is stochastically larger than $Y$, denoted $X \geq_{st} Y$, if for all $x$ we have $\mathbb{P}[X > x] \geq \mathbb{P}[Y > x]$. We use $\Phi$ and $\varphi$ to denote the standard Gaussian CDF and PDF respectively.

We treat prediction with expert advice as a special case of online linear optimization (Abernethy et al., 2014). At each time step $t \in [T]$, we choose an action $x_t$ from the action set $\mathcal{X}$, which in our case is the $n$-dimensional probability simplex. We then observe the gain vector $g_t \in \mathcal{G} \subseteq [0,1]^n$ and receive reward $\langle x_t, g_t \rangle$. *Static regret* is defined as the difference between the total reward we ultimately receive and the reward of the best single action in hindsight, i.e. $\max_{x \in \mathcal{X}} \sum_{t=1}^{T} \langle x, g_t \rangle - \sum_{t=1}^{T} \langle x_t, g_t \rangle$. Our utility analysis uses the *oblivious adversary* model, which means that gain vectors can take on arbitrary fixed values but can't depend on the realized values of the noise we add for privacy (but note that our LDP guarantees still hold even against adaptive adversaries (Denisov et al., 2022)).

---

[2]Shrinking Dartboard, by Geulen et al. (2010), a regularization-based antecedent to the perturbation-based algorithm of Devroye et al. (2013) that we build on.

---

**Algorithm 1** RW-FTPL(Devroye et al., 2013)

---

**Require:** Noise scale $\eta$, dimension $n$
   Sample $\tilde{G} \sim \mathcal{N}(0, \eta^2 I_n)$
   **for** $t = 1, \ldots, T$ **do**
      Play $x_t = \arg\max_{x \in \mathcal{X}} \langle x, \tilde{G} \rangle$
      Get $\tilde{g}_t \sim \mathcal{N}(g_t, \eta^2 I_n)$
      (Unobserved) receive reward $\langle x_t, g_t \rangle$
      Update $\tilde{G} \leftarrow \tilde{G} + \tilde{g}_t$

---

**Algorithm 2** RW-AdaBatch

---

**Require:** Scale $\eta$, dimension $n$, tolerance $\alpha$
   Sample $\tilde{G} \sim \mathcal{N}(0, \eta^2 I_n)$
   Initialize empty buffer, set delay to 0
   **for** $t = 1, \ldots, T$ **do**
      Play $x_t = \arg\max_{x \in \mathcal{X}} \langle x, \tilde{G} \rangle$
      Get $\tilde{g}_t \sim \mathcal{N}(g_t, \eta^2 I_n)$ and add to buffer
      (Unobserved) receive reward $\langle x_t, g_t \rangle$
      **if** delay is 0 **then**
         Move data from buffer to $\tilde{G}$
         Set new delay using Corollary 1
      **else**
         Decrement delay

---

To define privacy, we say that two sequences of gain vectors $g_1, g_2, \ldots, g_T$ and $g_1', g_2', \ldots, g_T'$ are *adjacent* if there is at most one index $i$ such that $g_i \neq g_i'$. A randomized mechanism $\mathcal{M}$ is said to satisfy *central* DP (CDP) (Dwork et al., 2014) if for for all adjacent sequence $g, g'$, the distributions of $\mathcal{M}(g_1, \ldots, g_T)$ and $\mathcal{M}(g_1', \ldots, g_T')$ satisfy a formal notion of indistinguishability which we define shortly. In the non-interactive (Smith et al., 2017) local (Yang et al., 2023) setting that we study, we instead imagine that $\mathcal{M}$ is applied separately to each data point as a preprocessing step, and our algorithm only gets access to the sequence $\mathcal{M}(g_1), \ldots, \mathcal{M}(g_T)$. Satisfying LDP then requires indistinguishability between the distributions of $\mathcal{M}(g_t)$ and $\mathcal{M}(g_t')$.

To define indistinguishability, we make use of $f$-DP (Dong et al., 2019), a modern DP variant based on hypothesis testing. Given two distributions $P, Q$, we define the tradeoff function $\mathcal{T}(P, Q) : [0, 1] \to [0, 1]$ so that $T(P, Q)(\alpha)$ is the minimum false negative rate achievable at false positive rate $\alpha$ when distinguishing $P$ from $Q$. A mechanism $\mathcal{M}$ operating on individual data points is then said to satisfy $f$-DP if $T(\mathcal{M}(g), \mathcal{M}(g')) \geq f$ for all $g, g' \in \mathcal{G}$. As a special case, we define the Gaussian tradeoff functions $G_\mu := \mathcal{T}(\mathcal{N}(0, 1), \mathcal{N}(\mu, 1))$ and say that a mechanism satisfies $\mu$-Gaussian DP ($\mu$-GDP) if it satisfies $G_\mu$-DP. This definition can be naturally satisfied by locally injecting Gaussian noise with scale calibrated to the sensitivity of our gain vectors, defined as $\Delta = \max_{g, g' \in \mathcal{G}} \|g - g'\|_2 \leq \sqrt{n}$, which is in fact what all of the algorithms we consider will do.

We also make occasional use of the traditional definition of approximate DP for comparison with prior work: a mechanism operating on individual data points is said to satisfy $(\varepsilon, \delta)$-DP if, for all $g, g' \in \mathcal{G}$ and all measureable events $S$, we have that $\mathbb{P}[\mathcal{M}(g) \in S] \leq e^\varepsilon \mathbb{P}[\mathcal{M}(g') \in S] + \delta$.

## 3   Prediction with Expert Advice Under LDP

The classical algorithm that inspired our approach was introduced by Devroye et al. (2013) under the name 'Prediction by random-walk perturbations.' We will refer to it as RW-FTPL for brevity. Given a symmetric distribution $\mathcal{D}$, the algorithm begins by sampling $\tilde{z}_0, \ldots, \tilde{z}_T \overset{iid}{\sim} \mathcal{N}(0, \eta^2 I_n)$. At each time step, it chooses $x_t = \arg\max_{i \in [n]} (G_{t-1} + S_{t-1})_i$, where $S_t := \sum_{s=0}^t z_t$ and $G_t = \sum_{s=0}^t g_s$. Here, $S_t$ is an element of a symmetric random walk, giving the algorithm its name.

When analyzing the privacy guarantees of the algorithm, it is helpful to view the decision rule from a different perspective. Define $\tilde{g}_t := g_t + z_t$, with $g_0 = 0$. RW-FTPL can be reformulated as a post-processing of these noisy gain vectors, which by our assumptions satisfy (local) GDP with privacy parameter $\mu = \Delta/\eta$. Using tools from convex analysis(Lee, 2018), it can also be shown that its expected static regret is at most $\left(\eta + \frac{2}{\eta}\right)\sqrt{2T \log n}$. In the absence of additional structure on $\mathcal{G}$, such as sparsity, this regret bound is weaker than the non-private baseline by a factor of $\frac{\sqrt{n}}{\mu}$. In recent work, Bhatt & Kostina (2025) prove matching lower bounds showing that this rate is asymptotically optimal when gain vectors are perturbed with additive Gaussian noise.

## 3.1 STABILITY UNDER LDP: RW-ADABATCH

The RW-FTPL algorithm was not originally designed with privacy in mind: the authors' goal was to create a low-regret algorithm which changes its prediction only a small number of times in expectation. Intuitively, this limited-switching means that RW-FTPL is *almost* invariant to permutations of nearby data points. RW-AdaBatch modifies RW-FTPL to be *truly* permutation-invariant over many data-dependent intervals. This gives data points that fall within large intervals a 'crowd' to hide in, amplifying privacy *a la* the shuffle model (Erlingsson et al., 2019) while still producing the same outputs with high probability. The amplification naturally grows stronger when data is 'easy' in the sense that $G_t$ has non-zero gap. Omitted proofs are in the appendix.

### 3.1.1 UTILITY ANALYSIS.

We show that the expected regret of RW-AdaBatch is at most $(1 + \alpha/2)$ times greater than the expected regret of RW-FTPL, where $\alpha$ is a small constant given to the algorithm as a parameter. Our main tool for proving this result is the following theorem on Gaussian random walks, which establishes conditions under which RW-FTPL's prediction will remain stable with high probability:

**Theorem 1.** *Let $x_0, x_1, \ldots, x_B \in \mathbb{R}^n$ be a Gaussian random walk with $x_0 = v$ and $x_{t+1} - x_t \sim \mathcal{N}(0, \eta^2 I_n)$. If $v$ has gap $k$, then the probability that the leader changes at any point during the random walk is at most $2\Phi(-\sqrt{2}\beta) + 2\sqrt{\pi}\varphi(-\beta)[\Phi(\beta) - \Phi(-\beta)]$, where $\beta = k/(\eta\sqrt{2B}) - \sqrt{\log(2n-2)}$. The same is true if $v$ has gap $k + \kappa$ and we wish to bound the probability that the gap ever dips below $\kappa$.*

**Corollary 1.** *The `ComputeDelay` subroutine presented in subsection A.6 selects the largest batch size $B_t$ such that the probability of RW-FTPL changing its prediction in the next $t$ time-steps can be bounded above by $\alpha\sqrt{\log n/(t + B_t)}$ using Theorem 1.*

**Corollary 2.** *For any $\alpha > 0$, if RW-AdaBatch chooses batch sizes using the `ComputeDelay` subroutine with parameter $\alpha$, then its expected regret is at most $(1 + \frac{\alpha}{2})(\eta + \frac{2}{\eta})\sqrt{2T \log n}$*

Corollary 1 follows because `ComputeDelay` directly solves the relevant optimization problem with a root-finding subroutine. Corollary 2 follows because the expected regret of RW-AdaBatch can be decomposed into the regret of RW-FTPL plus the additional regret incurred when batching leads the algorithms to make different predictions. Different predictions only occur when the leader changes within a batch, and so Corollary 1 lets us upper bound the regret from a particular batch over the interval $[t, t + B_t]$ by $B_t\alpha\sqrt{\log n/(t + B_t)}$. For each time step $\tau \in [t, t + B_t)$, this gives us amortized regret of $\alpha\sqrt{\log n/(t + B_t)} \leq \alpha\sqrt{\log n/\tau}$. Summing over all time steps in all batches shows that the extra expected regret over the entire input is at most $\sqrt{2}\alpha\sqrt{2T \log n}$, and adding the base regret of RW-FTPL along with the fact that $(\eta + \frac{2}{\eta}) \geq 2\sqrt{2}$ for $\eta > 0$ yields the corollary.

### 3.1.2 PRIVACY ANALYSIS.

RW-AdaBatch offers the same *local* privacy guarantee as RW-FTPL, i.e. $\mu = \Delta/\eta$. Ex-post, a point that falls in a batch of size $B$ enjoys a stronger *central*-DP guarantee of $\mu/\sqrt{B}$. To compute the ex-ante privacy amplification of RW-AdaBatch, we would like to exactly characterize the distribution of containing batch sizes any given point. This turns out to be intractable, and so we instead construct a proxy distribution and prove that it is stochastically smaller than the true containing batch size for a given point, independent of the data. The final privacy guarantees we derive have no closed form, but they can be efficiently computed; we visualize them in Figure 1.

The key reduction behind our argument is as follows: for any $t$ and $B$, there exists some threshold $\kappa$ such that RW-AdaBatch will always choose a batch of size greater than $B + 1$ if it observes a gap greater than $\kappa$ before time $t$. Let's say that a gap is 'small' if it is less than $\kappa$ and 'large' otherwise. We observe that whenever $g_t$ falls inside a batch of size $B + 1$ or less, the gap must have been small during at least one of the last $B$ time steps. By contrapositive, the gap being large at all of those time steps is a sufficient condition for $g_t$ to fall in a batch of size greater than $B + 1$. Crucially, this condition depends *only* on the distribution of the gaps, which makes it much easier to analyze than the full joint distribution over batch sizes. We lower bound the probability of this condition being satisfied by proving the following theorem, which is stated formally in the appendix:

**Theorem 2** (Informal)**.** *Let $K$ be the random variable representing the size of the gap at time $t$ in the execution of RW-AdaBatch. Then $K$ is stochastically smallest when $G_{t-1} = 0$.*

The proof of Theorem 2 involves studying integrals over level sets of the 'gap' function with respect to the Gaussian measure. When $n = 2$, these sets are convex and the proof is straightforward. For $n > 2$ the proof is highly non-trivial and so we defer it to the appendix. For our purposes, the important thing is that we can write down a formula for the CDF and PDF of the gap in this worst case scenario:

$$F_k(\varepsilon) = 1 - n \int_{-\infty}^{\infty} \varphi(x)\Phi(x - \varepsilon)^{n-1} \, dx \tag{1}$$

$$f_k(\varepsilon) = n(n-1) \int_{-\infty}^{\infty} \varphi(x)\varphi(x - \varepsilon)\Phi(x - \varepsilon)^{n-2} \, dx \tag{2}$$

This allows us to compute a lower bound the probability that the gap at time $t - B$ will be at least $k + \kappa$. We then return to Theorem 1 to upper bound the probability that the gap will drop below $\kappa$ within $B$ time steps given that it was initially $k + \kappa$. Integration over all possible values of $k$ then gives us a lower bound on the probability that the gap is at least $\kappa$ for all $B$ steps, from which we can derive a probability distribution which is provably stochastically smaller than the true containing batch size of $g_t$. From here, Lemma 1 lets us translate a stochastically smaller distribution on batch sizes into a lower bound on tradeoff functions, which is stated in terms of Gaussian mixture distributions. This tradeoff function lacks a simple closed form, but Lemma 2 allows us to evaluate it numerically.

**Lemma 1.** *Let $s_1 \sim P$ and $s_2 \sim Q$ for some distributions $P \geq_{st} Q$ over the non-negative real numbers. Then $\mathcal{T}\big((s_1, \mathcal{N}(0, s_1^2), (s_1, \mathcal{N}(1, s_1^2)\big) \geq \mathcal{T}\big((s_2, \mathcal{N}(0, s_2^2), (s_2, \mathcal{N}(1, s_2^2)\big).$*

**Lemma 2** (Joint Concavity of Tradeoff Functions (Wang et al., 2024))**.** *Let $P_w, Q_w$ be two mixture distributions, each with $m$ components and shared weights $w$. Then: $\mathcal{T}(P_w, Q_w)(\alpha(t, c)) \geq \sum_{b=1}^{m} w_b \mathcal{T}(P_b, Q_b)(\alpha_b(t, c)) =: \beta(t, c)$, where $\alpha_b(t, c) = \mathbb{P}_{X \sim P_b}\big[\frac{q_b}{p_b}(X) > t\big] + c\mathbb{P}_{X \sim P_b}\big[\frac{q_b}{p_b} = t\big]$ is the type 1 error of the log likelihood ratio test between $P_b$ and $Q_b$ with parameters $t$ and $c$, and $\alpha(t, c) = \sum_{b=1}^{m} w_b \alpha_b(t, c).$*

In our case, all of the components correspond to continuous distributions and so we can ignore the $c$ parameter. This characterization is sufficient to compute Figure 1, which visualizes the amplified tradeoff curves using the bounds proved above. In the appendix, we show how this can be losslessly converted into a curve of $(\varepsilon, \delta)$ guarantees using Proposition 2.13 from Dong et al. (2019).

## 3.2 PRIVATE LEARNING IN DYNAMIC ENVIRONMENTS: RW-META

Recall that *static regret* is defined with respect to the single best expert in hindsight. To make static regret guarantees more meaningful in dynamic environments, we would like to use experts that are themselves non-trivial learning algorithms, capable of adapting to trends in the private data. This is straightforward in the non-private context, where the term 'expert' is already understood to include such learners. The new challenge in the private setting is that, even though existing private algorithms are capable of privately *selecting* the best data-dependent expert, we could still violate privacy by *acting* on the chosen expert's advice unless the expert itself also satisfies DP.

Prior work (including the oblivious algorithm of Saha et al. (2025)) sidesteps this problem by only considering simple experts whose predictions are fixed independently of the data and therefore consume no privacy budget. A naïve alternative approach is to divide our privacy budget, allocating a small portion to each of $m$ data-dependent experts, but this will generally lead to each expert's error growing like $\sqrt{m}$. To overcome this dilemma, we propose RW-Meta, a new algorithm for privately selecting between *and* acting on the advice of data-dependent experts, *solely* through post-processing of the same noisy gain-vectors used by RW-FTPL. As a result, the error of each expert is constant with respect to $m$, improving significantly on the naïve baseline, and the algorithm as a whole satisfies $\mu$-GLDP by the post-processing invariance of DP Dwork et al. (2014).

### 3.2.1 UTILITY ANALYSIS

We refer to data-dependent experts as 'learners' and the algorithm selecting between them as the 'meta-learner.' We model learners as functions $f_1, \ldots, f_m : \mathbb{R}^{n,\infty} \to \mathcal{X}$. At time step $t$, each learner

---

**Algorithm 3** RW-Meta

---

**Require:** Noise scale $\eta$, learners $f_1, \ldots, f_m$
   Sample $\tilde{G}^{(m)} \sim \mathcal{N}(0, \eta^2 I_n)$; Initialize $\Sigma \leftarrow \eta^2 I_m$
   **for** $t = 1, \ldots, T$ **do**
      Set $\Sigma^* \leftarrow \Sigma - \frac{1}{m^2}(\vec{1}^T \Sigma \vec{1})\vec{1}\vec{1}^T$             # Decorrelate our base estimate of learner gains
      Set $\sigma^2 \leftarrow \max(2t, \lambda_{max}(\Sigma^*))$
      Sample $y_t \sim \mathcal{N}(0, \sigma^2 I - \Sigma^*)$
      Choose $j_t \leftarrow \arg\max_j (\tilde{G}^{(m)} + y_t)_j$         # Choose learner using the decorrelated estimate
      **for** $i \in [m]$ **do**
         $x_{i,t} \leftarrow f_i(\tilde{g}_0, \ldots, \tilde{g}_{t-1}) \in \mathbb{R}^n$
      Get $\tilde{g}_t \sim \mathcal{N}(g_t, \eta^2 I_n)$
      (Unobserved) receive reward $\langle x_{j_t,t}, g_t \rangle$
      Set $X_t \leftarrow [x_{1,t}, \ldots, x_{m,t}]^T \in \mathbb{R}^{m \times n}$
      Update $\tilde{G}^{(m)} \leftarrow \tilde{G}^{(m)} + X_t \tilde{g}_t$            # Update our base estimate
      Update $\Sigma \leftarrow \Sigma + \eta^2 X_t X_t^T$              # Record its new covariance matrix

---

suggests an action $x_{t,i} = f_i(\tilde{g}_1, \ldots, \tilde{g}_{t-1})$, and the meta-learner chooses $j_t \in [m]$ (by restricting learners to all rely on the same noisy gain vectors $\tilde{g}$, we avoid the issue of learners leaking sensitive information). The meta-learner then receives the gain associated with the action suggested by the chosen learner at this time step, i.e. $\langle x_{t,j_t}, g_t \rangle$. Our goal is to minimize the regret of the meta-learner with respect to the best single learner in hindsight.

The starting observation for our method is that the value $\langle x_{t,i}, \tilde{g}_t \rangle$ is an unbiased estimate for the gain of learner $i$ at time $t$. Specifically, let $X_t \in \mathbb{R}^{m \times n}$ be the matrix whose $i$th row is $x_{t,i}$. Then we have that $\sum_{s=1}^{t} X_t \tilde{g}_t \sim \mathcal{N}(G_t^{(m)}, \Sigma_t)$, where $G_t^{(m)} = \sum_{s=1}^{t} X_t g_t$ and $\Sigma_t = \eta^2 \sum_{s=1}^{t} X_t X_t^T$. So, like in RW-FTPL, our algorithm is able to maintain a Gaussian vector centered on the true gain of each learner, but with the inconvenient wrinkle that the covariance matrix is now data-dependent.

To dissolve this issue, we introduce a *decorrelation* step to the algorithm by defining a new matrix $\Sigma_t^* = \Sigma_t - \frac{1}{m^2}(\vec{1}^T \Sigma_t \vec{1})\vec{1}\vec{1}^T$. At each time step, we then sample a new Gaussian vector $y_t$ with mean zero and covariance matrix $\max(2t, \lambda_{max}(\Sigma_{t-1}^*))I - \Sigma_{t-1}^*$ and choose $j_t = \arg\max_j (\tilde{G}_{t-1}^{(m)} + y_t)_j$. Since our algorithm is invariant to additive noise with covariance $\vec{1}\vec{1}^T$, this lets us reduce to the scaled identity matrix covariance case, from which we derive the following theorem:

**Theorem 3.** *The expected regret of RW-Meta with respect to the best learner is at most*

$$\left[ \max \left( \sqrt{2},\ \eta \lambda_{max} \left( \frac{\Sigma_T^*}{\eta^2 T} \right)^{1/2} \right) + \sqrt{2} \right] \sqrt{2T \log m} = O\left( \frac{\sqrt{Tnm \log m}}{\mu} \right)$$

The best case scenario here is when all learners suggest different actions at each time step, in which case $\lambda_{max}(\Sigma_T^*/(\eta^2 T)) = 1$ and we obtain a tighter bound of $O(\frac{1}{\mu}\sqrt{Tn \log m})$. The worst case occurs when the learners are divided into cliques of size $m/2$. In both cases the regret bound is with respect to the best *learner*, which may be much better than the best *action*. Comparing the best-case bound to the regret of RW-FTPL illustrates that RW-FTPL can be seen as a special case of RW-Meta where $m = n$, all learners suggest different actions, and learner suggestions never change over time.

## 4   EMPIRICAL EVALUATION

**RW-AdaBatch.** On the basis of Theorem 2, we expect that the worst case for privacy occurs when all means are equal. To evaluate the privacy loss in this case, we simulate 1000 runs of RW-AdaBatch on a stream of all-zero data with $T = 10,000$. We then use the empirical PMF of containing batch sizes for each point to estimate its true tradeoff function. Our results are visualized in Figure 1.

Our results indicate that the analytic bounds are reasonably but not perfectly tight. The looseness primarily arises in the moderate FPR regime. This is because in bounding the probability that the

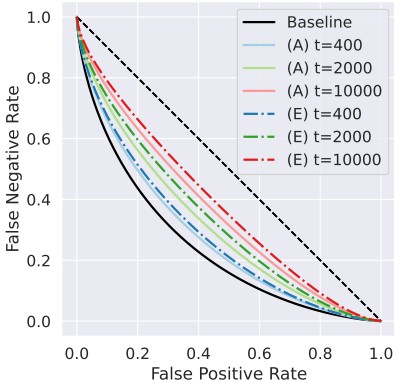 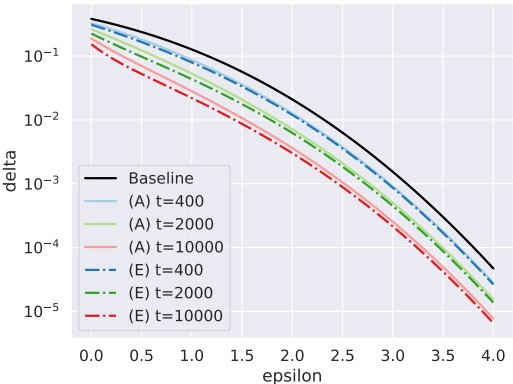

Figure 1: Equivalent representations of worst-case privacy amplification for RW-AdaBatch with parameters $\mu = 1$, $n = 25$, $\Delta = \sqrt{n}$, $\alpha = 0.01$. The baseline corresponds to the $G_\mu$ tradeoff curve, solid lines correspond to the **A**nalytic upper bound, and dash-dotted lines correspond to **E**mpirical Monte Carlo simulations. The analytic bound is fairly tight, particularly for small $\delta$/low FPR.

leader changes, we assume that the deterministic part of the gap will shrink at the fastest possible rate, but this is overly conservative in the oblivious adversary model. As a result, we underestimate the probability of selecting very large batch sizes. Meanwhile, the low FPR behavior of the algorithm is largely driven by the probability of seeing small batch sizes, where our bound is much tighter.

**RW-Meta.** We evaluate the performance of RW-Meta with real-world data from the COVID-19 pandemic. Specifically, we use data from the United States Department of Health and Human Services (DHHS, 2024)[3] containing weekly reports from thousands of hospitals from mid-2020 through 2023. For each state, our task is to predict which hospitals will report the highest density of COVID patients in each week. Because we are interested in preserving the privacy of individual patients and not the hospitals themselves, we modify the basic adjacency definition in subsection 2.2 to define two sequences as adjacent if they differ only in a single individual's participation at a single time step. For example, if every hospital in a state had at least 10 beds, then we would compute sensitivity for that week as $\sqrt{2}/10$, reflecting the possibility that someone might have not have gotten sick, *or* they might have chosen to go to a different hospital. In our dataset, the minimum number of beds ranges between 4.3 and 14.6, with the result that sensitivity is kept manageably low throughout.

Following Altieri et al. (2021), we consider models that first forecast the exact proportion of COVID cases in each hospital and then make a final prediction by taking the argmax of their forecast. The gain of a learner is the true proportion of COVID patients in the hospital they select. Our evaluation uses 13 learners consisting of rolling Gaussian regression algorithms with window sizes of 8/16/32/64 and weak/medium/strong regularization, as well as the basic RW-FTPL algorithm. We use Gaussian regression models because they are popular for medical forecasting and their theoretical assumptions match the additive Gaussian noise we use to protect privacy. The range of hyperparameters we consider was chosen manually through exploratory analysis with a disjoint slice of the dataset, guided by the goal of maximizing variety across learners. We evaluate our algorithm on data from three states (New Mexico, Pennsylvania, and California), which were chosen to cover a diverse range of population sizes while being geographically large enough to decorrelate different hospitals.

We remove any hospitals that did not appear to participate in the DHHS's data sharing program, defined as reporting fewer than 100 suspected cases over the 3 year period. This left us with data from 24, 146, and 293 hospitals in each state respectively, covering the 148 weeks between August 9th, 2020, and June 4th, 2023. We evaluate our algorithm under four different privacy levels: no privacy ($\mu = \infty$), low privacy ($\mu = 1$), medium privacy ($\mu = 0.5$) and high privacy ($\mu = 0.25$).

Because we are not aware of any existing LDP experts algorithms for oblivious adversaries, we compare RW-Meta against the state-of-the-art private experts algorithm of Agarwal & Singh (2017). This comparison must be caveated because their algorithm satisfies CDP and was not designed for dynamic environments; our goal is not to prove that one algorithm is superior, but rather to

---

[3]Also available at https://osf.io/nzt9x

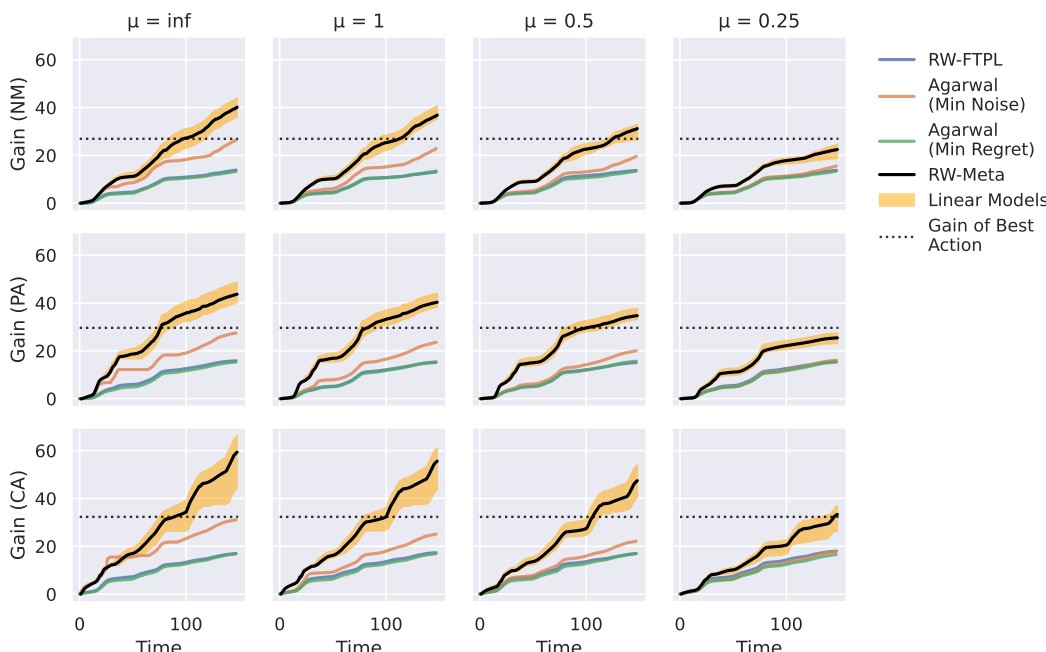

Figure 2: Results of empirical evaluation on COVID-19 hospitalization data, averaged over 100 iterations. Rows correspond to states while columns correspond to privacy levels. The shaded orange regions enclose the maximum and minimum gain of the 12 rolling regression learners, and the dotted black lines represent the maximum cumulative COVID density of any single hospital in the given state. Static regret can be interpreted as the distance between a learner's total gain and the dotted line, while the regret of RW-Meta corresponds to its distance from the top of the orange envelope. Note that all plots share the same $x$ and $y$ axis.

contextualize the costs of satisfying local vs. central DP and the potential benefits of moving beyond the paradigm of data-independent experts. We consider two choices of parameters for their algorithm: choosing the minimum noise scale required to still satisfy $\mu$-GCDP (Min Noise), and choosing the noise scale to optimize worst-case bounds on regret subject to privacy constraints (Min Regret).

**Results**. We repeat our evaluation 100 times and report average results, which are visualized in Figure 2 and summarized numerically in Table 2. Averages are reported alongside 95% confidence intervals based on the central limit theorem with Bonferroni correction. Across all settings, we find that RW-Meta consistently outperforms all other algorithms, followed by Min Noise. This is because RW-Meta is able to consistently achieve *negative* static regret at low and moderate privacy levels, which is essentially impossible to match using data-independent experts, regardless of noise scale.

Separately, we find that RW-Meta performs around 90% as well as the best linear model in each setting on average. Which exact linear model performs best varies considerably, however. For instance, the learner with window size 8 and strong regularization is the best performing learner in the high privacy setting for the Pennsylvania dataset, but only middle of the pack for New Mexico (where the best learner instead has a window size of 64). Across different privacy levels, the variation is even larger. This heterogeneity highlights the appeal of using meta-learning, as it eliminates the need to commit to a single set of parameters in advance.

**Conclusion.** We have presented two algorithms for the fundamental problem of prediction with expert advice, both of which satisfy LDP. In static environments, our RW-AdaBatch algorithm is a costless upgrade over the classical algorithm we build off of, improving the privacy of its outputs with provably insignificant impact on utility. In dynamic environments, our RW-Meta algorithm uses a novel, privacy-preserving technique to dynamically adapt to shifts in the input data, and we show that the resulting improvements in performance can be won at no additional privacy cost. Our theoretical results are supplemented by an empirical evaluation showing that our algorithms can achieve high performance in a real-world, privacy-critical prediction task.

Table 2: Average gain of learners

| Privacy Level | RW-Meta | Agarwal (Min Noise) | Agarwal (Min Regret) | Best Linear Model |
|---|---|---|---|---|
| New Mexico | | | | |
| $\mu = \infty$ | 40.14 $\pm$0.30 | 26.48 | 13.38 $\pm$0.38 | 43.91 |
| $\mu = 1$ | 36.87 $\pm$0.36 | 22.83 $\pm$0.37 | 13.41 $\pm$0.39 | 40.69 $\pm$0.26 |
| $\mu = 0.5$ | 31.19 $\pm$0.55 | 19.56 $\pm$0.42 | 13.25 $\pm$0.40 | 34.14 $\pm$0.49 |
| $\mu = 0.25$ | 22.46 $\pm$1.04 | 15.58 $\pm$0.43 | 13.38 $\pm$0.43 | 25.85 $\pm$1.26 |
| Pennsylvania | | | | |
| $\mu = \infty$ | 43.70 $\pm$0.41 | 27.42 | 15.51 $\pm$0.37 | 48.61 |
| $\mu = 1$ | 40.32 $\pm$0.50 | 23.55 $\pm$0.44 | 15.38 $\pm$0.37 | 44.30 $\pm$0.39 |
| $\mu = 0.5$ | 34.71 $\pm$0.68 | 20.03 $\pm$0.44 | 15.64 $\pm$0.40 | 38.06 $\pm$0.57 |
| $\mu = 0.25$ | 25.44 $\pm$0.87 | 16.09 $\pm$0.42 | 15.35 $\pm$0.38 | 28.73 $\pm$0.82 |
| California | | | | |
| $\mu = \infty$ | 59.36 $\pm$0.47 | 31.12 | 16.68 $\pm$0.36 | 66.23 |
| $\mu = 1$ | 55.62 $\pm$ 0.47 | 25.09 $\pm$ 0.51 | 16.91 $\pm$ 0.34 | 61.16 $\pm$0.29 |
| $\mu = 0.5$ | 47.43 $\pm$ 0.81 | 22.14 $\pm$ 0.48 | 16.82 $\pm$ 0.41 | 54.21 $\pm$0.56 |
| $\mu = 0.25$ | 33.34 $\pm$ 1.30 | 17.93 $\pm$ 0.41 | 16.84 $\pm$ 0.35 | 38.43 $\pm$1.30 |

## 4.1 REPRODUCIBILITY STATEMENT

Full proofs of all of our theoretical contributions are provided in the appendix, including clear statements of all necessary assumptions. For our empirical evaluation, we provide a reference to the dataset we use (DHHS, 2024), which is available at https://osf.io/nzt9x, and describe all data pre-processing steps in section 4. In subsection A.3, we describe the hardware and software used to carry out our experiments as well as specific implementation choices that improve the efficiency of our algorithms, and subsection A.6 provides explicit pseudo-code for the main subroutine of RW-AdaBatch. We provide containerized scripts for reproducing all of our experimental results at https://github.com/ben-jacobsen/dp-online-learning.

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

## A APPENDIX

### A.1 COMPLEXITY ANALYSIS

In the worst case, RW-AdaBatch requires us to find the root of a smooth monotonic function at each time step. Crucially, however, the size of the problem is constant and does not depend on $n$ or $T$, and so the asymptotic complexity of RW-AdaBatch is only $O(nT)$. We describe some of our practical techniques for improving the efficiency of this step in the following section.

RW-Meta algorithm requires $O(m^2 + mn)$ memory to store the predictions of the learners at each round as well as the $\Sigma$ matrix. Similarly, we require $O(m^2 + mn)$ operations per iteration to compute $\Sigma^*$ and $X_t \tilde{g}_t$. It is also necessary to choose a specific technique to compute $\lambda_{max}(\Sigma^*)$: in our implementation, we use the LOBPCG algorithm, which enjoys linear convergence and requires solving a $3 \times 3$ eigenproblem at each iteration (Knyazev, 2001). We can control the total number of iterations by warm-starting with the leading eigenvector of the previous iteration, which is guaranteed to be within $O(m)$ of the new maximum eigenvalue.

## A.2    Parameter Choice

All three of the algorithms we study require choosing a noise scale $\eta$. This can be done by computing the minimum noise scale required to guarantee privacy, i.e. $\eta = \Delta/\mu$, which typically works well on realistic data. To optimize worst-case performance for RW-FTPL or RW-AdaBatch, one can instead choose $\eta = \max(\sqrt{2}, \Delta/\mu)$.

In our early design process, we investigated several different choices of $\alpha$ for RW-AdaBatch and found that it seemed to have little practical impact on the behavior of RW-AdaBatch. This is not necessarily surprising — the random variables we are trying to control all have sub-Gaussian tails, resulting in failure probabilities that are exponentially tiny for most batch sizes. We take $\alpha = 0.01$ for simplicity in our experiments.

## A.3    Implementation Details

All experiments are carried out on a laptop running Ubuntu 22.04 with an intel i5-1135G7 CPU and 16GB of RAM. We implement our algorithms using Python 3.10, numpy 2.0.0 (Harris et al., 2020), scipy 1.14.0 (Virtanen et al., 2020), and mpmath 1.3.0 (mpmath development team, 2023).

Although RW-FTPL is computationally straightforward, Both RW-Meta and RW-AdaBatch include non-trivial computations as subroutines which must be implemented efficiently. We briefly describe our implementation choices here.

In the case of RW-Meta, we use the LOBPCG algorithm as implemented in scipy to find the leading eigenvalue of $\Sigma_t^*$ at each iteration, with the leading eigenvector from the previous round as an initial guess. We find that at the problem sizes we consider, this step is not a significant bottleneck (taking about 2ms per iteration), and we therefore do not pursue any further optimizations.

In the case of RW-AdaBatch, we take advantage of the fact that the bound from Theorem 1 is a monotonic function of $\beta$ by pre-computing many input/output pairs. To avoid issues with floating point precision when computing extreme values of the Gaussian PDF/CDF, we use the mpmath library for this task, which supports arbitrary precision arithmetic and numerical integration. We then interpolate the result with a monotonic cubic spline, which can be used to closely approximate the required value of $\beta$ necessary to achieve a given failure probability $\delta$. In practice, the discrete nature of the choice of batch size means that the small differences between the true inverse function and the interpolated approximation are insignificant, while the corresponding speed-up is dramatic.

Finally, prior work has shown that straightforward DP implementations are often vulnerable to attacks that take advantage of the idiosyncrasies of floating point numbers, leading to catastrophic privacy failures (Mironov, 2012). We therefore employ the secure random sampling method of Holohan & Braghin (2021) which renders these attacks computationally prohibitive.

## A.4    Limitations and Future Work

### A.4.1    Applicability of Local DP

The primary limitation of our work is that satisfying local DP sometimes requires adding unreasonably large amounts of noise, particularly when gain vectors are high-dimensional. To guarantee low regret, our approach requires gain vectors with sensitivity smaller than their literal dimension, or else a silo-like setting where several sensitive values can be averaged locally before being sent to the algorithm, but this may not always be achievable in practice.

### A.4.2    Extension to More Complex Learning Problems

While the experts problem is flexible enough to represent many real world tasks, it is also interesting from a theoretical perspective because it is one of the most fundamental problems in online learning. This naturally raises the question of whether the methods developed in this work can be extended to more complicated learning problems.

With respect to metalearning, RW-Meta fundamentally requires a linear gain function to compute unbiased estimates for the gain of each learner. So, while it can be extended to online linear

optimization problems beyond prediction with expert advice, it is not clear that the same method could be generalized to the larger class of online convex optimization.

Meanwhile, RW-AdaBatch relies on the property that the maximum of a linear function over a polyhedral set is always attained at one of those points. It would therefore be difficult to extend it to settings like online linear optimization over the sphere where the action set lacks this structure. On the other hand, the same property holds for maximums of convex functions over polyhedral sets, and so it is possible that the ideas behind RW-AdaBatch could be extended to some instances of online convex optimization.

### A.5 Proof of Theorem 1

The formal statement of Theorem 1 is as follows:

*Let $x_0, x_1, \ldots, x_B \in \mathbb{R}^n$ be a Gaussian random walk with $x_0 = v$ and $x_{t+1} - x_t \sim \mathcal{N}(0, \eta^2 I_n)$. If $v$ has gap $k$, then the probability that the leader changes at any point during the random walk is at most $2\Phi(-\sqrt{2}\beta) + 2\sqrt{\pi}\varphi(-\beta)\big[\Phi(\beta) - \Phi(-\beta)\big]$, where $\beta = k/(\eta\sqrt{2B}) - \sqrt{\log(2n-2)}$. The same is true if $v$ has gap $k + \kappa$ and we wish to bound the probability that the gap ever dips below $\kappa$.*

The proof of the theorem requires the following fundamental results about the extrema of Gaussian processes:

**Lemma 3.** *Let $z_1, \ldots, z_B \sim \mathcal{N}(0, \eta^2)$, and let $S_t = \sum_{i=1}^{t} z_i$. Then:*

$$\max_t S_t \leq_{st} |S_B| \tag{3}$$

*Proof.* Instead of considering the discrete random walk, we consider the continuous analogue in one-dimensional Brownian motion over $[0, B]$ with scale $\eta$. The maximum value attained by the discrete random walk is at most as large as the maximum value attained by the Brownian motion. The latter is known to follow a half-normal distribution with scale $\eta\sqrt{B}$ (see e.g. Section 3 of Majumdar et al. (2020)), which is also the distribution of $|S_B|$. $\square$

We additionally need the well-known Borell-TIS inequality, which states that the maximum of Gaussians concentrates closely around its expected value (Adler & Taylor, 2007):

**Lemma 4** (Borell-TIS). *Let $\{f_t\}$ be a centered Gaussian process on $T$. Denote $\|f\| = \sup_{t \in T} f_t$ and $\sigma_T = \sup_{t \in T} \sigma_t$. Then for any $u \geq 0$,*

$$\mathbb{P}(\|f\| > \mathbb{E}[\|f\|] + u) \leq \exp\left(\frac{-u^2}{2\sigma_T^2}\right) \tag{4}$$

Finally, we have the following standard upper-bound on the expected maximum of Gaussians based on moment-generating functions:

**Lemma 5.** *Let $z_1, \ldots, z_n$ be (not-necessarily independent) random variables such that $z_i \sim \mathcal{N}(0, \sigma_i^2)$. Let $Z = \max_i z_i$. Then:*

$$\mathbb{E}[Z] \leq \max_i \sigma_i \sqrt{2 \log n} \tag{5}$$

*Proof.* We have that:

$$
\begin{aligned}
\exp(t\mathbb{E}[Z]) &\leq \mathbb{E}[e^{tZ}] \\
&\leq \sum_{i \in [n]} \mathbb{E}[e^{tz_i}] \\
&= \sum_{i \in [n]} \exp(\sigma_i^2 t^2 / 2) \\
&\leq n \exp(\max_i \sigma_i^2 t^2 / 2)
\end{aligned}
$$

Taking log of both sides and choosing $t$ to minimize the upper bound gives the desired result. $\square$

Now: suppose that at a given time step, the gap of $\tilde{G}$ is $k$. Our goal is to bound the probability that our algorithm's output will change in the next $B$ time steps. Because the algorithm is maximizing a linear objective function, it will always output one of the vertices of the probability simplex. Without loss of generality, assume that the current time step is 0 and that the current leader has index 1, and write $S_{t,i} = \sum_{s=1}^{t}(\tilde{g}_t)_i$. Then the following is a necessary condition for our algorithm's prediction to change:

$$\max_{j>1} \max_{t\in[B]} S_{t,j} > k + \min_{t\in[B]} S_{t,1} \tag{6}$$

Lemma 3 tells us that each maximum over $t$ is stochastically smaller than a half-normal distribution with scale $\eta\sqrt{B}$, and by symmetry the minimum on the right hand side is stochastically larger than a negative half-normal. We can write a half-normal random variable as the maximum of a Gaussian random variable and its negation, and so Lemma 5 lets us bound the expectation of the left-hand side with $\eta\sqrt{2B\log(2n-2)}$. Denote this quantity as $E$. Then, assuming that $k > E$, we can use Lemma 4 along with the independence of each coordinate to upper bound the probability of our event with:

$$\int_{-\infty}^{0} p(\min_t S_{t,1} = z)\mathbb{P}(\max_{t,j>1} S_{t,j} > k + z)\,dz$$
$$= 2\Phi\Big(\frac{E-k}{\eta\sqrt{B}}\Big) + 2\int_0^{k-E} \frac{1}{\eta\sqrt{B}}\varphi\Big(\frac{E-k+u}{\eta\sqrt{B}}\Big)\exp\Big(\frac{-u^2}{2B\eta^2}\Big)\,du$$

Here, the first term represents the probability that the current leader dips below the expected maximum value of the $n-1$ remaining actions, and the second term represents the probability that one of those $n-1$ actions manages to overtake the current leader regardless. Using the definition of the Gaussian PDF, we can rewrite the second term as:

$$\exp\Big(\frac{-(E-k)^2}{4B\eta^2}\Big)\int_0^{k-E} \frac{1}{\eta\sqrt{B}}\varphi\Big(\frac{2u+(E-k)}{\eta\sqrt{2B}}\Big)\,du$$
$$= \frac{1}{\sqrt{2}}\exp\Big(\frac{-(E-k)^2}{4B\eta^2}\Big)\int_{(E-k)/(\eta\sqrt{2B})}^{(k-E)/(\eta\sqrt{2B})} \varphi(y)\,dy$$
$$= \frac{1}{\sqrt{2}}\exp\Big(\frac{-(E-k)^2}{4B\eta^2}\Big)\Big[\Phi\Big(\frac{k-E}{\eta\sqrt{2B}}\Big) - \Phi\Big(\frac{E-k}{\eta\sqrt{2B}}\Big)\Big]$$

Write $\beta = (k-E)/(\eta\sqrt{2B})$. Then the entire upper bound can be simplified as:

$$2\Phi(-\sqrt{2}\beta) + 2\sqrt{\pi}\varphi(-\beta)\big[\Phi(\beta) - \Phi(-\beta)\big] \tag{7}$$

which proves the theorem. $\square$

### A.6 COMPUTEDELAY SUBROUTINE

This appendix contains the explicit computation used by RW-AdaBatch to compute the size of the next batch:

---

**Algorithm 4** Subroutine to compute delays

---

**Require:** Noise scale $\eta$, gap $k$, dimension $n$, tolerance $\alpha$

$E \leftarrow \sqrt{\log(2n-2)}$

$U_1(B) := 2\Phi\Big(\frac{B-k}{\eta\sqrt{B}} + \sqrt{2}E\Big)$

$U_2(B) := 2\sqrt{\pi}\varphi\Big(\frac{k-B}{\eta\sqrt{2B}} - E\Big)$

$U_3(B) := \Phi\Big(\frac{k-B}{\eta\sqrt{2B}} - E\Big) - \Phi\Big(\frac{B-k}{\eta\sqrt{2B}} + E\Big)$

$\delta_t(B) := \alpha\sqrt{\frac{\log n}{t+B}}$

$B \leftarrow \texttt{FindRoot}\big(U_1(B) + U_2(B)U_3(B) - \delta_t(B)\big)$

**return** $\max(0, \texttt{Floor}(B))$

---

A.7 PROOF OF THEOREM 2

The formal statement of Theorem 2 is as follows:

*Let $S_\varepsilon = \{v \in \mathbb{R}^n : v_{(n)} - v_{(n-1)} \leq \varepsilon\}$ and let $\gamma$ denote the standard Gaussian measure on $\mathbb{R}$. Then for any vector $\mu \in \mathbb{R}^n$, $\gamma^n(S_\varepsilon) \geq \gamma^n(S_\varepsilon - \mu)$.*

The set $S_\varepsilon$ is invariant under permutations and shifts in the $\vec{1}$ direction, and so we can assume without loss of generality that $\mu_1 \geq \mu_2 \geq \ldots \geq \mu_n = 0$. It is easiest to begin by finding the measure of the complement of our set, which is a union of disjoint sets where one element is the leader and the gap is greater than $\varepsilon$. This measure is given by:

$$f(\mu) = \gamma^n((S_\varepsilon - \mu)^C)$$
$$= \sum_{i=1}^n \int_{-\infty}^\infty \varphi(x - \mu_i) \prod_{j \neq i} \Phi(x - \varepsilon - \mu_j) \, dx$$

Our strategy is to show that the partial derivative of $f$ with respect to $\mu_1$ is non-negative. To do so, we can use the fact that each of our disjoint sets is invariant under shifts in the $\vec{1}$ direction, and so the gradient of each summand with respect to $\mu$ must be orthogonal to $\vec{1}$. So, we can write our partial derivative as:

$$\sum_{i=2}^n \int_{-\infty}^\infty \varphi(x - \mu_1)\varphi(x - \varepsilon - \mu_i) \cdot \prod_{j \neq 1,i} \Phi(x - \varepsilon - \mu_j)dx$$
$$- \sum_{i=2}^n \int_{-\infty}^\infty \varphi(x - \varepsilon - \mu_1)\varphi(x - \mu_i) \cdot \prod_{j \neq 1,i} \Phi(x - \varepsilon - \mu_j)dx$$

Combining like terms, this gives us:

$$\sum_{i=2}^n \int_{-\infty}^\infty \varphi(x - \mu_1)\varphi(x - \mu_i) \, \exp\left(\varepsilon x - \frac{1}{2}\varepsilon^2\right)$$
$$\cdot \left[\exp(-\varepsilon\mu_i) - \exp(-\varepsilon\mu_1)\right] \prod_{j \neq 1,i} \Phi(x - \varepsilon - \mu_j) \, dx$$

Then, since $\mu_1 \geq \mu_i$ and $\varepsilon > 0$, the term inside the brackets is non-negative. Therefore, as a sum of integrals of products of non-negative values, the whole expression is non-negative.

From here, the fact that $S_\varepsilon$ is closed under permutations means that, if $\mu_1 = \mu_i$ for some $i$, then they have the same partial derivative. So, we can explicitly construct a path from any arbitrary $\mu$ to the origin along which $f(\mu)$ is non-increasing: first reduce $\mu_1$ until it equals $\mu_2$, then reduce both until they equal $\mu_3$, and so on. This construction shows that the function is globally optimized at $\mu = 0$, as desired. $\square$

**Remark 1.** *In the special case where $n = 2$, $S_\varepsilon$ is a convex set and the statement can be proved directly using Anderson's theorem (Anderson, 1955) or its extension by Marshall and Olkin for Schur-concave sets (Marshall & Olkin, 1974). When $n > 2$, however, the set is neither convex nor Schur-concave, necessitating the more explicit analysis here.*

**Remark 2.** *The informal statement of the theorem in the main text follows from this result together with the assumption of an oblivious adversary. The fact that the adversary is oblivious corresponds to the order of quantifiers in the formal theorem statement: first we choose an arbitrary vector $\mu$, and only afterwards do we observe the size of the gap. This is the only context in which the adversarial model is relevant for privacy — all of our LDP guarantees hold in both models (Denisov et al., 2022).*

A.8 PROOF OF LEMMA 1

The proof follows from a theorem by Blackwell (1951), reproduced as Theorem 2.10 in Dong et al. (2019), which states that $T(P, Q) \leq T(P', Q')$ iff there exists a randomized algorithm proc such that $\text{proc}(P) = P'$ and $\text{proc}(Q) = Q'$. We will construct such an algorithm.

Suppose we receive the random input $(s_2, x) \sim Q \times \mathcal{N}(b, s_2^2)$ for some a priori unknown $b$. We first choose $s_1 = F_P^{-1}(F_Q(s_2))$. Since $s_2 \sim Q$, we have that $F_Q(s_2) \sim \mathsf{Unif}(0, 1)$ and therefore $s_1 \sim P$. Then, since $P \geq_{st} Q$, we have that $F_P^{-1} \geq F_Q^{-1}$ and therefore $s_1 \geq s_2$. So, we can sample $z \sim \mathcal{N}(0, s_1^2 - s_2^2)$ and release the tuple $(s_1, x + z) \sim P \times \mathcal{N}(b, s_1^2)$, which completes the proof. $\square$

## A.9 CONVERSION TO $\varepsilon, \delta$-DP

We prove the following corollary to the privacy analysis in subsection 3.1:

**Corollary 3.** *For all $\varepsilon > 0$, RW-AdaBatch satisfies $(\varepsilon, 1 - e^\varepsilon \alpha(\varepsilon) - \beta(\varepsilon))$-DP, with $\alpha$ and $\beta$ defined as in Lemma 2.*

The proof relies on the Proposition 2.13 from Dong et al. (2019), which we reproduce here:

**Lemma 6** (Primal to Dual). *Let $f$ be a symmetric trade-off function. A mechanism is $f$-DP if and only if it is $(\varepsilon, \delta(\varepsilon))$-DP for all $\varepsilon \geq 0$ with $\delta(\varepsilon) = 1 + f^*(-e^\varepsilon)$, where:*

$$f^*(y) = \sup_{-\infty < x < \infty} yx - f(x) \tag{8}$$

*is the convex conjugate of $f$.*

In our case, $f$ is only defined on $[0, 1]$, so we let $f(x) = \infty$ for $x \notin [0, 1]$ and the supremum is effectively taken over $0 \leq x \leq 1$.

To find the convex conjugate, we need to find the specific value $\alpha(t)$ that optimizes $y\alpha(t) - \beta(t)$ for a given $y$. To that end, define $h_y(\alpha) = y\alpha - \mathcal{T}(\mathcal{A}(D), \mathcal{A}(D'))(\alpha)$. Then we have that $f^*(y) = h_y(\alpha^*)$, where $\alpha^* = \inf\{\alpha \in [0, 1] : 0 \in \partial h_y(\alpha)\}$.

We have that $\alpha(t)$ is differentiable with respect to $t$, so we can compute that:

$$\frac{d}{dt} h_y(\alpha(t)) = \frac{d}{dt}\left(y\alpha(t) - \sum_{b=1}^m w_b \beta_b(t)\right)$$

$$= y \sum_{b=1}^m w_b \frac{d}{dt}\alpha_b(t) - \sum_{b=1}^m w_b \frac{d}{dt}\beta_b(t)$$

$$= y \sum_{b=1}^m w_b\left(\frac{-1\sqrt{b}}{\mu}\varphi\left(\frac{t\sqrt{b}}{\mu} + \frac{\mu}{2\sqrt{b}}\right)\right)$$

$$- \sum_{b=1}^m w_b \frac{\sqrt{b}}{\mu}\varphi\left(\frac{t\sqrt{b}}{\mu} - \frac{\mu}{2\sqrt{b}}\right)$$

Define:

$$z_b = \frac{t\sqrt{b}}{\mu} - \frac{\mu}{2\sqrt{b}}$$

Then the expression simplifies to:

$$\frac{-1}{\mu}\left(\sum_{b=1}^m \sqrt{b}w_b y\varphi(z_b + \mu/\sqrt{b}) + \sum_{b=1}^m \varphi(z_b)\right)$$

$$= \frac{-1}{\mu}\sum_{b=1}^m \sqrt{b}w_b\varphi(z_b)\left[1 + y\exp\left(\frac{-\mu}{\sqrt{b}}z_b - \frac{\mu^2}{2b}\right)\right]$$

$$= \frac{-1}{\mu}\sum_{b=1}^m \sqrt{b}w_b\varphi(z_b)\left(1 + y\exp(-t)\right)$$

Setting this equal to 0 and using the fact that $\varphi, w, b > 0$, we obtain that:

$$y\exp(-t) + 1 = 0$$
$$t = \ln(-y)$$

Denote this last quantity as $t_y$. From this it follows that $f^*(y) = h_y(\alpha(t_y))$, and therefore that our mechanism satisfies $\varepsilon, \delta(\varepsilon)$-DP for all $\varepsilon > 0$ and:

$$\delta(\varepsilon) = 1 - e^\varepsilon \alpha(t_{-e^\varepsilon}) - \beta(t_{-e^\varepsilon}) = 1 - e^\varepsilon \alpha(\varepsilon) - \beta(\varepsilon)$$

as desired. $\qquad\qquad\qquad\qquad\qquad\qquad\qquad\qquad\qquad\qquad\qquad\qquad\qquad\qquad\qquad\qquad$ $\square$

### A.10 PROOF OF THEOREM 3

We formally analyze the regret of RW-Meta using the convex analysis framework of Lee (2018). Let $M(G) = \max_{x \in \mathcal{X}} \langle x, G \rangle$ be the baseline potential function, and for any set of distributions $\{\mathcal{D}_t\}$, define the smoothed potential function $\tilde{M}_t(G) = \mathbb{E}_{z \sim \mathcal{D}_t} M(G + z)$. Finding the expected regret of any randomized FTPL-style algorithm can be reduced to finding the regret of the deterministic algorithm which plays $\mathbb{E}_{z \sim \mathcal{D}_t}[\arg\max_{x \in \mathcal{X}} M(G + z)] = \nabla \tilde{M}_t$ at each time step $t$, which by Lemma 3.4 in Lee (2018) satisfies the following equality:

$$\mathsf{Regret} = \sum_{t=1}^{T} \Big( \underbrace{\big( \tilde{M}_t(G_{t-1}) - \tilde{M}_{t-1}(G_{t-1}) \big)}_{\textit{overestimation penalty}} + \underbrace{D_{\tilde{M}_t}(G_t, G_{t-1})}_{\textit{divergence penalty}} \Big) + \underbrace{M(G_T) - \tilde{M}_T(G_T)}_{\textit{underestimation penalty}}$$

Where $D_f(y, x) = f(y) - f(x) - \langle \nabla f(x), y - x \rangle$ is the *Bregman divergence*, which is a measure of how quickly the gradient of $f$ changes. Intuitively, the overestimation penalty represents the error that results when we fool ourselves into believing an action is better than it really is by adding too much noise, and the divergence penalty represents the error that comes from always playing one step behind the real objective function. Because we add zero-mean noise, the underestimation penalty is always negative by Jensen's inequality, and so to prove low regret it suffices to upper bound the first two terms.

For the overestimation penalty, we can use the convexity of $M$ to bound:

$$\tilde{M}_t(G_{t-1}) - \tilde{M}_{t-1}(G_{t-1})$$
$$= \mathbb{E}_{z \sim \mathcal{N}(0,1)}[M(G_{t-1} + \Sigma_t^{1/2} z) - M(G_{t-1} + \Sigma_{t-1}^{1/2} z)]$$
$$\leq \mathbb{E}_{z \sim \mathcal{N}(0,1)}[M((\Sigma_t^{1/2} - \Sigma_{t-1}^{1/2})z)]$$

From which it follows by telescoping that the entire sum is upper bounded by:

$$\mathbb{E}_{z \sim \mathcal{N}(0,1)}[\Sigma_T^{1/2} z]$$
$$\leq \sqrt{2 \log(m) \max(\eta^2 T, \lambda_{max}(\Sigma_T^*))}$$

Where the last step follows from Lemma 5. For the divergence penalty, we use Lemma 3.14 in Lee (2018) to bound:

$$D_{\tilde{M}_t}(G_t, G_{t-1}) \leq \frac{\sqrt{2 \log m}}{\eta_t}$$

From which it follows that the sum can be upper bounded by $\sqrt{2 \log m} \cdot \sum_{t=1}^{T} \frac{1}{\sqrt{2t}} \leq 2\sqrt{T \log m}$, giving us our final regret bound of:

$$\left[ \max\left( \sqrt{2}, \eta \cdot \lambda_{max}\left(\frac{\Sigma_T^*}{\eta^2 T}\right)^{1/2} \right) + \sqrt{2} \right] \sqrt{2T \log m} \tag{9}$$

### A.11 HEURISTIC PRIVACY ANALYSIS OF RW-ADABATCH

The mixture-distribution method of privacy analysis developed in Wang et al. (2024) and used in the analysis of RW-AdaBatch leads to very tight bounds, but this comes at the cost of being highly complex and difficult to interpret. To help build some intuition, this appendix derives a heuristic estimate for the asymptotic level of privacy amplification provided by RW-AdaBatch. We make the following simplifying assumptions:

1. The variance of the noisy cumulative gain vector is very large, i.e. $t\eta^2 \gg 1$, and so the deterministic component of the gap is insignificant compared to the stochastic component. This allows us to simplify our equations by dividing everything through by $\eta$, and so we will also assume WLOG that $\eta = 1$.

2. Potential batch sizes are relatively small, i.e. $B_t \ll t$, and so we can simplify the target failure probability $\delta_t = \alpha\sqrt{\log n/(t + B_t)} \sim \alpha\sqrt{\log n/t}$.

3. We approximate the distribution of *containing* batch sizes for time $t$ with the distribution of batch sizes induced by the gap *at* time $t$. We believe this is a reasonable heuristic for sufficiently large $t$ (intuitively, if the gap is large at time $t$ and Assumptions 1 and 2 hold, then we expect the gap at the preceding few time steps to be essentially the same size), and it allows us to sidestep much of the complexity in the derivation of our rigorous lower bounds on privacy amplification presented in subsection 3.1.

4. Initially, we will assume that $n = 2$, and therefore that the gap itself follows a half-normal distribution. After establishing bounds in this setting, we'll return to consider the case of general $n$.

Under these assumptions, consider the event that $t$ falls in a batch of size less than $B + 1$. Under Assumption 3, this is equivalent to the gap at time $t$ being smaller than the threshold computed by RW-AdaBatch when it selects batch sizes. We will compute that threshold. In the notation of subsection A.5, we have $E = \sqrt{2B\log(2n-2)}$, and by Assumption 2 we have $\delta_t = \alpha\sqrt{\log n/t}$. Plugging these quantities into Lemma 4 and letting $k$ represent the size of the gap at time $t$, we get:

$$\exp\left(\frac{-(k-E)^2}{2B}\right) > \alpha\sqrt{\log n/t} \tag{10}$$

$$(k-E)^2 < 2B\left(\log(1/\alpha) + \frac{1}{2}\log(t/\log n)\right) \tag{11}$$

$$k < E + \sqrt{B} \cdot \sqrt{2\log(1/\alpha) + \log(t/\log n)} \tag{12}$$

Next, under Assumptions 1 and 4, we have that $k = |z|$, where $z \sim \mathcal{N}(0, 2t)$. So, we can upper-bound the probability that our event occurs with the small-ball inequality $\mathbb{P}[|z/\sqrt{2t}| \leq a] \leq \frac{2a}{\sqrt{2\pi}}$, giving us:

$$\mathbb{P}[\text{batch size } < B+1] = \mathbb{P}\left[|z/\sqrt{2t}| < \sqrt{\frac{B}{2t}}\left(\sqrt{2\log(2n-2)} + \sqrt{2\log(1/\alpha) + \log(t/\log n)}\right)\right] \tag{13}$$

$$\leq \sqrt{\frac{B}{\pi t}}\left(\sqrt{2\log(2n-2)} + \sqrt{2\log(1/\alpha) + \log(t/\log n)}\right) \tag{14}$$

Finally, if a point falls in a batch of size $B + 1$, then its post-hoc privacy level is $\mu' := \mu/\sqrt{B+1} < \mu/\sqrt{B}$. This means that the expression above can also be interpreted as an upper-bound on the probability that the amplification ratio $r := \mu/\mu'$ for point $t$ is less than $\sqrt{B}$. Rearranging, we can derive the following bound for the amplification ratio that holds with probability at least $1 - \gamma$:

$$r = \frac{\gamma^2 \pi t}{\left(\sqrt{2\log(2n-2)} + \sqrt{2\log(1/\alpha) + \log(t/\log n)}\right)^2} \tag{15}$$

$$= O\left(\frac{\gamma^2 t}{\log(1/\alpha) + \log(t)}\right) \tag{16}$$

And so in the special case where $n = 2$, we see that the quantiles of the privacy amplification provided by RW-AdaBatch grow like $t/\log(t)$. Importantly, however, this bound is only pointwise and not uniform over all quantiles simultaneously.

The case of general $n > 2$ is more complicated because the gap no longer follows a nice, analytically-tractable distribution. However, we can use the fact that $\Phi(x) \leq 1$ to derive a very loose upper-bound on Equation 1 by:

$$F_k(\varepsilon; n) = 1 - n \int_{-\infty}^{\infty} \varphi(x)\Phi(x - \varepsilon)^{n-1}\, dx \tag{17}$$

$$\geq 1 - n \int_{-\infty}^{\infty} \varphi(x)\Phi(x - \varepsilon)\, dx \tag{18}$$

$$= 1 - \frac{n}{2}(1 - F_k(\varepsilon; 2)) \tag{19}$$

This allows us to convert a $1 - \gamma$ bound on any monotonic function of the gap when $n = 2$ into a $1 - n\gamma/2$ bound when $n > 2$. Plugging this into the bound on the amplification ratio above, we get:

$$r = O\Big(\frac{\gamma^2 t}{n^2(\log n + \log(1/\alpha) + \log t)}\Big) \tag{20}$$

Showing that the degree of amplification remains almost linear in $t$, but potentially with a slower rate that depends on the number of experts.

### A.11.1 Limitations of Heuristic Analysis

Although we believe this analysis is helpful for building intuition, we caution against using it as a replacement for the more rigorous bounds derived in subsection 3.1.

Firstly, this is because Assumptions 1-3 discard several factors which are asymptotically irrelevant but potentially meaningful at realistic problem sizes. Assumption 3 in particular is difficult to rigorously justify, although it seems to hold reasonably well in practice.

Secondly, the bounds derived above are only meaningful if $\gamma = \omega(1/\sqrt{T})$. This is an issue because there will still often be a meaningful risk of selecting a very small batch size even when $t$ is fairly large — in particular, this is almost certain to happen every time the leader changes. So while it might be possible to make heuristic statements like "With probability about 0.9, your post-hoc privacy level will be at least as good as $\mu = 0.1$", the risk that we'll get unlucky and observe $\mu = 1$ instead makes this sort of guarantee meaningfully weaker than actually offering a privacy guarantee of $\mu = 0.1$ from the beginning.

In the language of $f$-DP, our heuristic should convey a fairly good idea for what the moderate-FPR region of the tradeoff curve looks like while giving an inflated impression of the privacy guarantee in the low-FPR regime. This can be contrasted with the true lower bounds we compute analytically, which are over-pessimistic in the moderate-FPR regime but quite accurate in the (often-more-critical) regime when $\alpha$ is small.

