# OpenReview forum: "Prediction with Expert Advice under Local Differential Privacy"
_ICLR.cc/2026/Conference — ICLR 2026 Poster_

### Official Review · Reviewer_M68M · 2025-10-21

**Soundness:** 3
**Presentation:** 2
**Contribution:** 3
**Rating:** 4
**Confidence:** 3

**Summary:**

This paper studies the problem of online prediction with expert advice under local DP constraints. It introduces two algorithms, RW-AdaBatch and RW-Meta, and provides theoretical analyses of the proposed methods. The paper also includes experimental results on a real-world dataset.

**Strengths:**

- The paper studies a relevant problem.
- The proposed methods are supported by formal regret bounds and empirical experiments on real-world data.

**Weaknesses:**

- From Table 1, the proposed algorithms appear to improve over RW-FTPL only by a small constant factor, which does not seem significant.
- The paper lacks a regret lower bound for the problem.

**Questions:**

- In Line 151, the authors state that the LDP guarantees still hold even against adaptive adversaries. Does the algorithm also remain valid in the adaptive setting? If so, what is the utility in this case?
- Why is there no experimental comparison with Asi et al. (2023)? From Line 103, the authors mention that Asi et al. (2023) studies the high-dimensional regime. Could you further what this means and why it makes direct comparison difficult?

---

> ### Author Response · Authors · 2025-11-19
>
> Thank you for your helpful review! We address each of the questions and weaknesses you raise below:
>
> **Weakness 1**: We believe that the degree of improvement is best illustrated by the figures in section 4.  For instance, figure 1 demonstrates that the privacy amplification provided by RW-AdaBatch is quite significant, corresponding to e.g. a nearly 10x improvement in the value of $\delta$ when $t=10,000$, and we achieve this with worst-case regret bounds that are only 0.5% weaker than RW-FTPL's. That's a dramatically better privacy-utility tradeoff than you could achieve by modifying the noise-scale used by RW-FTPL and using the standard analysis of the Gaussian mechanism.
>
> Similarly, the regret definition of RW-Meta is qualitatively stronger than that of RW-FTPL because it is competing against a stronger class of experts, and Figure 2 and Table 2 show that this translates to actual gain that is $2-3\times$ what RW-FTPL can achieve. That is, our improvement comes from achieving similar regret guarantees with respect to a stronger baseline, rather than by achieving stronger regret guarantees with respect to the same baseline. We consider the introduction of this stronger baseline to itself be an important contribution, even if it cannot be expressed quantitatively in the same way as e.g. shaving off a factor of $\sqrt{\log n}$ while studying an established problem.
>
> **Weakness 2**: We agree that a general lower bound for the experts problem under LDP would be a very interesting theoretical advance.  In our specific setting of non-interactive LDP,  we note that the asymptotic regret of $O(\frac{1}{\mu}\sqrt{Tn\log n}$ achieved by RW-FTPL matches lower bounds on $\ell_\infty$ error for the simpler task of simply estimating the sum of all $T$ gain vectors (Duchi et al. 2014, Proposition 4). Intuitively, an algorithm that cannot reliably distinguish an expert with gain $G$ and an expert with gain $G + O(\frac{1}{\mu} \sqrt{Tn \log n})$ should expect to suffer $O(\frac{1}{\mu} \sqrt{Tn \log n})$ regret, and so we suspect that RW-FTPL is asymptotically optimal, but this is only a heuristic argument. More generally, we feel that the area of prediction with expert advice under LDP remains extremely underexplored, and we hope that our paper will inspire more people to work on these sorts of interesting open questions.
>
> **Question 1**: This is an excellent question! Our utility guarantees do in fact rely critically on the oblivious adversary assumption. The key issue here is that in non-interactive LDP, we’re forced to sample random noise once for each data point and commit to that randomness for the rest of the interaction. An adaptive adversary would therefore be comparable to an adversary with access to the internal random coins of our algorithm, and such adversaries are impossible to compete with.
>
> **Question 2**: During the process of conducting our evaluation, we did actually implement the Private Shrinking Dartboard algorithm of Asi et al. (2023). However, it performed extremely poorly on our dataset. On closer investigation, we found that this is because their algorithm has a very long “warm up” period during which its outputs are chosen almost uniformly at random, which is a tradeoff that they accepted because it allowed them to achieve a very strong asymptotic dependency on the number of experts.
>
> After some consideration, we decided that it would be unfair to evaluate their algorithm on an input that didn’t match the problem setting that they were targeting (i.e. one with a very long time horizon and large number of experts). On the other hand, the algorithm of Agarwal and Singh (2017) is technically much more similar to our own work: it also uses additive Gaussian noise, making it easier to compare privacy guarantees, and it works quite well even when the time horizon is relatively short. So, we decided that their algorithm made the most sense as a CDP baseline.

---

> > ### Author Response · Authors · 2025-11-26
> > **Ping**
> >
> > Hello,
> >
> > The discussion period ends in a week, and we'd like to make sure that we've adequately addressed your concerns. Please let us know if you have any further questions, or if there's anything that you would like clarified!

---

> > > ### Comment · Reviewer_M68M · 2025-11-28
> > >
> > > Thank you for the response. My remaining concern is that the improvements shown in Table 1 are only constant-factor in theoretical terms, and without a regret lower bound, the overall theoretical significance of the paper remains unclear to me. The empirical results are promising, but for a primarily theoretical paper, it would be helpful to see a more explicit theoretical separation or a matching lower bound.

---

### Official Review · Reviewer_1cZk · 2025-10-23

**Soundness:** 2
**Presentation:** 2
**Contribution:** 3
**Rating:** 4
**Confidence:** 3

**Summary:**

The work studied the problem of prediction from expert advice under local differential privacy constraints. Building on random-walk follow-the-perturbed-leader (RW-FTPL), the authors first showed that the perturbation in RW-FTPL can imply LDP guarantee. Based on it, they designed the RW-AdaBatch algorithm to achieve privacy amplification. And they also designed RW-Meta algorithm for dynamic environments by adopting a noise-sharing scheme. They provided stability,  theoretical privacy and utility analysis for RW-FTPL and regret guarantee for RW-Meta. Finally, they also provided empirical results on the COVID-19 task to show the advantages of their methods.

**Strengths:**

1. The paper investigated an interesting problem of prediction from expert advice in online learning under differential privacy.
2. The paper provided some theoretical analysis and guarantees.
3. The authors implement their methods on real data.

**Weaknesses:**

1. The statement of writing and notation is confusing. For example, the paper mixed the statements of LDP and CDP. In the title, the authors used LDP, but in many places, they claim CDP. For the privacy parameter, somewhere $\epsilon$ is used, and somewhere $\mu$ is used.
2. Lacks direct numerical comparisons with recent LDP online learning works [1].
3. See more in the Questions part.

[1]. Cheng, Duo, Xingyu Zhou, and Bo Ji. "Follow-the-Perturbed-Leader for Adversarial Bandits: Heavy Tails, Robustness, and Privacy."

**Questions:**

1. Why is the description for RW-FTPL from lines 188-192 different from Algorithm 1?
2. The final step in Algorithm 2: set a new delay using Theorem 1. It is unclear to me how to set it. Can you explain more about it?
3. What does "Ada"  mean in "RW-AdaBatch"? Could you please give some explanation for it?
4.  What is the advantage of RW-AdaBatch compared with RW-FTPL? And from your Table 1, the regret of RW-AdaBatch is worse than RW-FTPL where $\alpha>0$?

---

> ### Author Response · Authors · 2025-11-20
>
> Thank you for your very helpful and detailed review! We respond to each of your questions and concerns individually below:
>
> **Weakness 1:** We are sorry that this caused confusion! We wish to clarify that references to CDP are intentional and not a notational mixup. The privacy amplification provided by RW-AdaBatch is conceptually similar to the idea of privacy amplification by shuffling, which essentially converts LDP guarantees into (quantitatively stronger) CDP guarantees. In other words, there’s one algorithm with two distinct privacy guarantees - a baseline LDP guarantee that holds against the shuffler (i.e. the entity running RW-AdaBatch), and a stronger CDP guarantee that holds against everybody else.
>
> This is noteworthy because in general, if all you know is that an algorithm satisfies LDP with certain privacy parameters, then it’s completely possible that it only satisfies CDP with exactly the same parameters and no better. For example, we don’t know of any way to prove that the outputs of RW-FTPL satisfy a CDP guarantee stronger than $\mu$-GDP. But, we can prove that the outputs of RW-AdaBatch are significantly more private than that, and that the level of privacy it provides grows as time passes or when the data is `easy’ in the sense of having a clear best expert.
>
> Your point that this can cause confusion is well-taken, however, and we’ve modified our presentation of RW-AdaBatch in the introduction to explicitly flag that privacy amplification by shuffling is a form of `local-to-central’ amplification. We’ve also added a statement to the end of Section 2.2 to clarify that we use $\varepsilon, \delta$ for the specific purpose of comparison with prior work, while $f$-DP is the definition of indistinguishability that we will use in our own analysis.
>
> **Weakness 2:** Thank you for bringing this paper to our attention! We were not previously aware of it, but on review, it isn’t clear to us that it attempts to solve the same problem that we do. In particular, it appears to be focused entirely on the bandit setting, whereas all of our algorithms assume full-information. We could certainly cite it when we discuss other prior work for private online learning with partial information, but the only available version of the paper we were able to find is hosted on OpenReview. If you’re aware of a published version of the paper (or a version hosted on a pre-print server like arXiv), would you be able to share the exact reference with us?
>
> **Question 1:** We describe RW-FTPL in two different ways like this because we feel that both perspectives are useful. If you're  analyzing the utility guarantees of a FTPL-style algorithm, then it's easiest to think of $\tilde{G}$ as a sum of the true cumulative gain and a single random variable which is drawn from a Gaussian random walk (this is how Devroye et al. present their algorithm, for instance, and it’s also why all of our algorithms have ‘Random Walk’ in their name). On the other hand, if you want to analyze the LDP guarantee, then it's more helpful to change your perspective and think of $\tilde{G}$ as a sum of many noisy gain vectors. We use both perspectives at various points in our paper and want to make that clear from the start.
>
> **Question 2:** This process is described in Appendix 1.6 under the title 'ComputeDelay Subroutine'. At a high level, the idea is to express the probability that the leader will change within the next $B+1$ steps as a function of $B$, and then use a root-finding algorithm to select the largest integer value of $B$ such that that probability isn’t too large compared to the tolerance parameter $\alpha$. The solution to this optimization problem then becomes the new batch size. To make this more clear, we’ve revised the presentation of the algorithm to reference the corollary to Theorem 1 that describes the ComputeDelay subroutine, instead of referencing Theorem 1 itself.
>
> **Question 3:** Ada stands for 'Adaptive', as in 'Adaptive Batching'. The name is chosen as a reference to AdaGrad, which is a classic optimization algorithm that uses an adaptive learning rate based on the data seen so far.
>
> **Question 4:** We discuss this in our response to Weakness 1, but briefly: the advantage of RW-AdaBatch is that its outputs enjoy much stronger privacy guarantees than RW-FTPL's. This is illustrated in Figure 1, which compares the privacy guarantee of RW-FTPL (Baseline) with the guarantees offered by RW-AdaBatch. The latter steadily grow over time as more data is gathered and it becomes possible to construct larger batches.
>
> This improvement in privacy comes at the cost of a very small multiplicative increase in worst-case regret bounds, but in this case the privacy-utility tradeoff is extremely favorable - for $t=10000$, we can enjoy a nearly $10\times$ improvement in the value of $\delta$ in exchange for 0.5% weaker regret bounds, which is much better than the privacy-utility tradeoff from the standard analysis of the Gaussian mechanism.

---

> > ### Author Response · Authors · 2025-11-26
> > **Ping**
> >
> > Hello,
> >
> > The discussion period ends in a week, and we'd like to make sure that we've adequately addressed your concerns. Please let us know if you have any further questions, or if there's anything that you would like clarified!

---

### Official Review · Reviewer_J3Gx · 2025-10-31

**Soundness:** 3
**Presentation:** 2
**Contribution:** 3
**Rating:** 4
**Confidence:** 3

**Summary:**

This paper studies the classical problem of prediction with expert advice under local differential privacy (LDP) constraints. The authors show that an existing algorithm naturally satisfies LDP. Moreover, the authors extend this algorithm to two new ones, named RW-AdaBatch and RW-Meta. RW-AdaBatch leverages the limited switching behavior of the existing algorithm to achieve adaptive privacy amplification while incurring a minimal cost to utility. RW-Meta extends the classical algorithm to data-dependent experts, which themselves are online learning algorithms. Finally, the authors complement their theoretical results with experiments on real-world data reported by hospitals during the COVID-19 pandemic.

**Strengths:**

- The paper studies an important problem at the intersection of privacy and online learning
- The theoretical results seem to be novel, and the experimental results are strong.

**Weaknesses:**

My biggest issue with this paper is its lack of clarity regarding the significance of its results. While I don't doubt that results and proofs are novel, I am having a hard time seeing what these results improve upon in private online learning.

For example, a regret bound of $\frac{\sqrt{T}}{\eps}$  under local DP can be derived from Theorem 4.1 in [1] by doing the same analysis except using the full-information multiplicative weights algorithm instead of EXP2.  In light of this, what are the main contributions of your work? Do you obtain a better privacy-utility tradeoff than this? Is your algorithm more efficient than running MWU on the noisy loss vectors? Do your algorithms obtain better central privacy-utility tradeoffs? More generally, why should I run your algorithm over just MWU on the noisy loss vectors?

In addition, as there are no closed-form privacy guarantees, it was not clear to me how the proposed algorithms will be practically useful. That is, how should a practitioner translate their desired privacy guarantee to the correct hyperparameters in the proposed algorithms? It would be great if the authors could walk me through how to do this.

[1] Agarwal, Naman, and Karan Singh. "The price of differential privacy for online learning." International Conference on Machine Learning. PMLR, 2017.

**Questions:**

See weaknesses.

---

> ### Author Response · Authors · 2025-11-19
>
> Thank you for your very helpful and constructive review! We address each of the concerns you raise individually below:
>
> **Comparison with MWU on noisy [gain] vectors**: Yes, all of the algorithms that we consider achieve better privacy-utility tradeoffs than running MWU on noisy gain vectors.
>
> For RW-FTPL (which we emphasize is a baseline and _not_ an original contribution), one advantage of FTPL approaches in private online learning is that the noise we add to protect privacy pulls double duty as implicit regularization. In an LDP context, adding additional regularization by running MWU on top of the noisy vectors would likely over-regularize the problem. This might recover the same asymptotics, but in practice it would be a weaker baseline than RW-FTPL. That said, we believe that the connection between RW-FTPL and Agarwal and Singh 4.1 is noteworthy, and have included a reference to it in the revised introduction.
>
> For RW-AdaBatch, we do achieve a better central privacy-utility tradeoff than both RW-FTPL and MWU over noisy gain vectors. In general, any privacy analysis of an online learning algorithm based on post-processing of noisy gain vectors can only be used to show a CDP guarantee equal to the LDP guarantee. Our use of batching makes it possible to analyze privacy at the level of consecutive subsequences of vectors, which leads to provably stronger CDP guarantees that are very nearly free: with the choice of $\alpha = 0.01$, they correspond to worst-case bounds that are only 0.5% weaker than RW-FTPL’s, and in practice we have never actually observed an instance where batching led to degraded performance. We try to communicate this distinction through our use of bold letters in Table 1, as well as various points in the introduction and conclusion, e.g. ``RW-AdaBatch algorithm is a costless upgrade over the classical algorithm we build off of, improving the privacy of its outputs with provably insignificant impact on utility.”
>
> For RW-Meta, we achieve a better privacy-utility tradeoff than both RW-FTPL and MWU over noisy gain vectors because RW-Meta is capable of competing against a stronger class of experts than prior works have considered, i.e. experts that base their predictions off of the private data seen so far. To better quantify this improvement, we’ve rewritten the preamble to Section 3.2 to explicitly contrast RW-Meta with the obvious baseline of dividing the privacy budget $m$ ways between the all data-dependent experts. That approach requires adding noise with scale $O(\sqrt{m}/\mu)$ to each learner’s input, causing the performance of each learner to degrade significantly as $m$ increases. In contrast, RW-Meta only requires adding $O(1/\mu)$ noise to each learner’s input while matching the same $\sqrt{m}$ dependency on regret in the worst-case (and doing significantly better when learners are well-behaved).
>
> **Setting hyperparameters**: To ensure that any of our algorithms satisfy $\mu$-GLDP, it is sufficient to select $\eta = \Delta/\mu$. But, suppose that a practitioner specifically wanted to satisfy a property like ``For any $t \geq t_0$, RW-AdaBatch will satisfy $\varepsilon, \delta$-CDP with respect to the replacement of $g_t$.” For any _fixed_ choice of parameters $\alpha, \Delta, \eta$, they could check whether it would satisfy their desired privacy guarantee as follows:
>
> - For every candidate batch size $B+1$, binary search with the ComputeDelay subroutine in Appendix A.6 to find a value $\kappa$ sufficient to guarantee that $g_t$ will fall into a batch of size at least $B+1$ if the gap is at least $\kappa$ all throughout the interval $[t-B, t]$.
> - Next, use Equation 1 to compute a lower-bound probability that the gap at time $t-B$ is at least $\kappa + k$ for any $k > 0$.
> - From here, Theorem 1 gives a lower bound on the probability of the gap ever dipping below $\kappa$ given that it starts at $\kappa + k$.
> - Combining the previous steps and numerically integrating over $k > 0$ then yields a lower bound on the probability that $g_t$ is contained in a batch of size at least $B+1$ (and therefore have an ex-post CDP guarantee of $\mu / \sqrt{B+1}$). Combining for all $B$ gives us a lower bound on the CDF of containing batch sizes.
> - Plugging the probabilities from our estimated CDF into Lemma 2 gives us a tradeoff function that we can numerically evaluate and check against the desired privacy level, e.g. by using Corollary 3 in the appendix.
>
> The process described above is how we produced Figure 1. In particular, we will release scripts to perform this computation as open-source software once the double-blind review period has ended. To use this pipeline for hyperparameter selection, the only adjustment necessary would be to wrap everything with a binary search over values of $\eta$.

---

> > ### Author Response · Authors · 2025-11-26
> > **Ping**
> >
> > Hello,
> >
> > The discussion period ends in a week, and we'd like to make sure that we've adequately addressed your concerns. Please let us know if you have any further questions, or if there's anything that you would like clarified!

---

> > > ### Comment · Reviewer_J3Gx · 2025-11-26
> > >
> > > I thank the authors for their detailed reply. I have increased my score, but I highly recommend that the authors include some discussion of the comparison to running MWU on the noisy gain vectors in the final version. I found the authors response pretty helpful and think other readers would benefit from this. Same goes for the hyper parameter selection.

---

### Official Review · Reviewer_7KQP · 2025-11-02

**Soundness:** 3
**Presentation:** 2
**Contribution:** 2
**Rating:** 4
**Confidence:** 2

**Summary:**

This paper studies prediction with expert advice under local differential privacy. That is, a learner aims to minimize the regret when the gains provided are noisy. It introduces two algorithms RW-AdaBatch, which modifies Random-Walk Follow-the-Perturbed-Leader (RW-FTPL) to adaptively batch inputs when predictions are unlikely to change, and RW-Meta, a meta-learner that privately selects among data-dependent experts using shared Gaussian noise.

**Strengths:**

- Extends the prediction-with-expert-advice framework to the local DP regime.
- RW-AdaBatch builds on RW-FTPL, showing that when its predictions stay stable, several updates can be grouped together. This  effectively mixes them like shuffling does in batch privacy, hence, gains privacy at little extra cost.
- Introduces a new LDP setting where the meta-learner selects among data-dependent experts sharing the same privatized data stream.
- In RW-Meta, the idea of reusing noise across learners is interesting.

**Weaknesses:**

- The work’s strength is theoretical rather than practical. Local-DP assumptions are hard to satisfy in realistic online settings, and the empirical evaluation is limited in scale and scope.
- The novelty seems to be low. RW-AdaBatch’s batching idea is a reformulation of existing privacy-amplification and RW-Meta reuses known post-processing principles rather than introducing a fundamentally new privacy mechanism.
- RW-Meta’s design involves maintaining and updating full covariance matrices across all learners, which is feasible for small ensembles but would scale poorly if the number of learners grows, limiting practical flexibility.
- The COVID-19 hospital experiment is limited (three states, one task), lacks baselines under equal conditions, and provides little sense of how privacy actually trades off against accuracy.
- The paper is dense and somewhat hard to follow. Intuitions such as why batching helps or how covariance correction works are found in the proofs rather than illustrated clearly in the main paper.

**Questions:**

1) RW-AdaBatch’s privacy amplification is demonstrated analytically but not empirically. Have the authors simulated or estimated the effective $\epsilon$-$\delta$ improvement to validate the theoretical claim?

1) RW-Meta assumes that all learners observe the exact same privatized data stream. In more realistic scenarios, learners might instead see partially correlated noisy versions of the same underlying data. Could the authors comment on whether their analysis could extend to this setting, and how such correlated noise would affect both privacy and regret guarantees?

2) RW-Meta maintains and updates full covariance matrices across learners. Up to what number of learners $m$ is this tractable in practice? What are realistic values of $m$? Could approximate or sparse updates mitigate the cost?

3) The evaluation focuses solely on a COVID-19 hospital dataset with one prediction task, which feels somewhat artificial. Can the authors explain why this setup meaningfully represents real online decision problems under local privacy, or discuss how their methods might generalize to more natural data streams?

---

> ### Author Response · Authors · 2025-11-19
>
> Thank you for your thorough and constructive review! We address your questions and concerns one-by-one below:
>
> **Weakness 1:** Could you expand on which specific assumption of LDP you consider unrealistic? From a trust perspective, LDP is strictly easier to satisfy than CDP because it eliminates the need for a central trusted curator, and the only thing we require local data-owners to compute is Gaussian noise. We view this ease of use as a meaningful practical advantage over alternatives like CDP or cryptographic aggregation methods.
>
> **Weakness 2:** RW-AdaBatch does not reformulate existing privacy-amplification: all prior works on shuffling are in the _batch_ setting, where the order of the data has no inherent meaning. A priori, it is not at all clear that something similar should even be possible in the _online_ setting, where the order of the data is critical. The techniques that we use to prove privacy amplification are novel, and the specific dynamics that we prove this amplification follows are particular to the online setting that we study.
>
> Similarly, RW-Meta also introduces new technical tools and concepts that do not exist in prior work. For instance, we don’t know of any earlier works that clearly state the conceptual distinction between data-dependent and data-independent experts, but the difference is critical in practical scenarios where we want to actually act on the advice of the expert we choose. Finally, we emphasize that our primary technical contribution with RW-Meta lies in the analysis of its utility guarantees and not its privacy analysis.
>
> **Weakness 3:** This is definitely a reasonable concern, but we believe that it’s unlikely to cause issues in practice: in the worst case, the regret bound of RW-Meta can scale with $O(\sqrt{m})$, and so there are already very compelling practical reasons to want to keep $m$ relatively low even before introducing computational constraints.
>
> **Weakness 4:** We are not certain that we understand what you mean by ‘lacking baselines under equal conditions’ in this context. Would you be able to expand on which specific aspect of the experimental design you are concerned about?
>
> With respect to your second point, we believe Section 4 already shows how privacy trades off against accuracy.  The columns of Figure 2 correspond to varying levels of privacy, and the $y$ axes of all plots are identical, so you can see how accuracy varies with privacy by visually comparing the heights of the lines. Similarly, Table 2 reports average utility across varying privacy levels for each dataset and algorithm that we evaluate. That said, we are certainly open to suggestions for how this tradeoff could be made clearer.
>
> **Weakness 5:** This is helpful feedback, thank you! To fix this, we have expanded the introduction to Section 3.1 to provide more intuition for why batching is helpful (i.e. because it provides points with a large `crowd’ to hide in), and we have rewritten our preamble of Section 3.2 to hopefully be more intuitively clear.
>
> **Question 1:** We **do** evaluate the privacy amplification of RW-AdaBatch empirically. This is described in the first two paragraphs of Section 4, and our results are visualized in Figure 1 (with curves based on empirical experiments labeled with an E).
>
> **Question 2:** In the context of RW-Meta, the learners are algorithms that we ourselves get to choose and have complete control over. We therefore believe that it *is* realistic to assume that they all receive the same noisy data stream: in the applications of RW-Meta that we envision, all of them would run on the same computer.
>
> **Question 3:** See weakness 3.
>
> **Question 4:** We believe that public health is a very well-motivated setting for LDP. On the one hand, there are clear social benefits to pooling information from many healthcare providers. On the other hand, there are often many regulatory and ethical restrictions on how the health data can be shared. LDP is a promising alternative to the more heuristic solutions in use today, like noiseless aggregation and suppression of small values, which lack formal privacy guarantees.
>
> Our specific forecasting task is motivated by the idea of a non-profit that can distribute medical supplies and wants to anticipate which hospital will have the greatest need for them in the near future. We operationalize ‘greatest need’ by the proportion of beds filled by Covid patients because this aligns with the goal of `flattening the curve’ by keeping hospitals under capacity. The dataset that we use is borrowed from the medical forecasting literature because most works on private online learning only evaluate their algorithms on synthetic data (if at all), and so there aren’t any standardized benchmarks in wide use. Finally, we note that our methods can also generalize to any other measure of utility which is bounded and low sensitivity, as well as to other instances of Online Linear Optimization, which we discuss briefly in Appendix A.4.2.

---

> > ### Author Response · Authors · 2025-11-26
> > **Ping**
> >
> > Hello,
> >
> > The discussion period ends in a week, and we'd like to make sure that we've adequately addressed your concerns. Please let us know if you have any further questions, or if there's anything that you would like clarified!

---

### Meta-Review · Area_Chair_XoTS · 2025-12-26

**Summary:**

- The paper starts with an interesting observation about the privacy guarantee that comes for free with RW-FTPL (Devroye et al 2013). This observation leads to a new algorithm, RW-Adabatch, that improves privacy-utility trade-off by playing on batch additions that are delayed in order to increase the stability of the experts.
- Because following the  advice of data-dependent experts can violate privacy, choosing an advice between multiple experts can improve privacy. The authors take inspiration on privacy amplification by shuffling to prove the privacy amplification guarantees in dynamic environments.
- The paper has new proof techniques but it applies on low dimension settings and has limited scalability on the number of experts. The improvement with respect to RW-FTPL is somehow limited (constant) and there is no regret lower bound

**Reviewer Concerns:**

- The concerns of the three first reviewers were correctly addressed (Clarity, positioning, limited experiments, scalability, ...) without changing the content of the article, but clearly arguing any disagreements with the reviewers or clarifying the position.
- Only the answers to M68M are less convincing.

**Reviewer Scores:**

I think that the first 3 reviewers could have increased their score. But the first one has low confidence compared to M68M.

I later realized that  J3Gx raised his score from 4 to 6, M68M lower his score from 6 to 4 but before the rebuttal,

---

### Decision · Program_Chairs · 2026-01-26

Accept (Poster)